# Mimicking efferent nerves using a graphdiyne-based artificial synapse with multiple ion diffusion dynamics

Huanhuan Wei[1], Rongchao Shi[1], Lin Sun[1], Haiyang Yu[1], Jiangdong Gong[1], Chao Liu[1], Zhipeng Xu[1], Yao Ni[1], Jialiang Xu [1✉] & Wentao Xu [1✉]

A graphdiyne-based artificial synapse (GAS), exhibiting intrinsic short-term plasticity, has been proposed to mimic biological signal transmission behavior. The impulse response of the GAS has been reduced to several millivolts with competitive femtowatt-level consumption, exceeding the biological level by orders of magnitude. Most importantly, the GAS is capable of parallelly processing signals transmitted from multiple pre-neurons and therefore realizing dynamic logic and spatiotemporal rules. It is also found that the GAS is thermally stable (at 353 K) and environmentally stable (in a relative humidity up to 35%). Our artificial efferent nerve, connecting the GAS with artificial muscles, has been demonstrated to complete the information integration of pre-neurons and the information output of motor neurons, which is advantageous for coalescing multiple sensory feedbacks and reacting to events. Our synaptic element has potential applications in bioinspired peripheral nervous systems of soft electronics, neurorobotics, and biohybrid systems of brain–computer interfaces.

[1] Institute of Photoelectronic Thin Film Devices and Technology, Key Laboratory of Optoelectronic Thin Film Devices and Technology of Tianjin, College of Electrical Information and Optical Engineering, School of Materials Science and Engineering, National Institute for Advanced Materials, Nankai University, Tianjin, P. R. China. ✉email: jialiang.xu@nankai.edu.cn; wentao@nankai.edu.cn

C omplex human nervous systems, which have the advantages of being highly compact, parallel, and reliable, are gaining increasing attention in various fields, such as neuromorphic computing, bioinspired sensorimotor systems, brain–machine interfaces, and prosthetics[1–10]. Somatosensory nerves transfer signals through synaptic connections to achieve a variety of perceptions, memories, and motion outputs with different depths[11–14]. Therefore, the imitation of synapses as building blocks for neural processing is of crucial importance for constructing efficient artificial sensorimotor systems, biohybrid systems, and neuromorphic chips[15–24]. For emulating synaptic behavior, a variety of structures, such as metal/insulator/ metal (MIM)-stack switches[25–29], electrolyte/semiconductor heterojunctions[30,31], and multiterminal transistors[32–36], have been employed to realize signal transmission. Among them, the configurations featuring ion migration can opportunely facilitate the emulation of biological sensory/motor neurons by neuromorphic synapses in bioinspired ionotronic systems and biohybrid systems[9,18,19,37]. However, exploration of the working principle of new materials and responsivity of devices, such as the thermal stability and environmental stability, is still imperative for achieving biomimetic functionalities.

Highly conjugated π-extended graphdiyne (GDY) has emerged as a new carbon allotrope[37–39] that serves to enable efficient batteries[40], catalysts[41], solar cells[42], nonlinear optics[43], electronic devices[44], and biomedical applications[45–47] owing to its remarkable optoelectronic properties and biocompatibility[45,48–53]. In particular, the interesting network of GDY with moderate triangular pores and $sp$-hybridized carbon atoms provides storage sites and rapid diffusion channels for alkali metal ions and even perchlorate ions[49–51,54]. Meanwhile, the relatively low diffusion barrier for ions in GDY contributes to its surface adsorption and interlayer insertion[55–57]. These intriguing ion shuttle characteristics inspire a new idea of constructing GDY-based artificial synapses (GASs) for mimicking synaptic cleft information transmission, which is promising for plasticity-mediated signal processing and transmission in biohybrid systems and artificial sensorimotor systems.

Inspired by biological motor neurons, an integrated ionic artificial efferent nerve can be constructed to mimic the real-time processing and manipulation of signals. Here, for the first time, a junction-type GAS is proposed by coupling a GDY film with solid-state electrolytes to emulate multiple short-term plasticity such as postsynaptic current, paired-pulse facilitation (PPF), and dynamic filtering, with outstanding pulse responsiveness and femtowatt-level energy consumption. The GASs can retain good ion diffusion dynamics under relatively high temperature (~353 K) and humidity (~35%). Attempts to exploit the short-term plasticity of GDY in a bioinspired analogous efferent nerve demonstrate real-time information integration, parallel processing capabilities, and signal transduction and actuation and therefore pave the way for future bioinspired ionotronic sensorimotor systems.

## Results

### General concept
Imitating the principle of the biological synaptic cleft, a GAS with a junction structure of electrolyte/GDY was fabricated to emulate the essential plasticity (Fig. 1). Communication between cells hinges on the propagation of action potentials along axons (Fig. 1a), and sodium ions ($Na^+$) eventually flow into the postsynaptic membrane with the flow of calcium ions ($Ca^{2+}$) when the action potential reaches the front of the junction (Fig. 1b). Electrolytes with lithium ($Li^+$) and $Na^+$ are exploited to prepare GASs, referred to as Li-GAS and Na-GAS, respectively. When a set of positive pulses (several millivolts

to several volts) are applied to the top of the artificial synapse (Fig. 1c), the alkali metal ions ($Li^+$ and $Na^+$) are forced to migrate to the gap and accumulate on the surface or even insert into the interlayer, coinciding with the charging process of rechargeable batteries (Fig. 1d)[54]. The investigation of the kinetics of ions at the interface facilitates the modulation of the synaptic response. Furthermore, signal integration and transduction can be realized by connecting several inputs of the GAS with artificial muscles and outputting the action response of different curvatures on artificial efferent neurons.

### Fabrication and operation of GASs
Two-dimensional (2D) GDY is produced from the catalytic coupling reaction of its precursor hexakis[(trimethylsilyl)ethynyl]benzene (HEB-TMS) through the liquid/liquid interfacial protocol (Supplementary Fig. 1a)[38,39]. An atomic force microscopy (AFM) image of the as-prepared GDY reveals its 2D nanoflake morphology with a thickness of approximately 3.3 nm (Fig. 2a). Layered structures with highly wrinkled nanosheets of GDY are observed under scanning electron microscopy (SEM, Supplementary Fig. 1b). Transmission electron microscopy (TEM) images (Fig. 2b, c) of a GDY nanoflake clearly demonstrate its lattice fringes with a spacing of approximately 0.45 nm, indicating the high crystallinity of the prepared GDY samples[43]. The selected area electron diffraction (SAED) pattern further illustrates the good crystallinity of the fabricated GDY (Fig. 2d). The high-resolution X-ray photoelectron spectroscopy (XPS) spectra show the deconvoluted C 1 s peaks with major contributions from $C \equiv C$ and $C = C$ species, indicating the $sp$- and $sp^2$-hybridized carbon atoms of GDY (Fig. 2e). The proportion of $sp/sp^2$ carbon close to 1.5 is also consistent with the chemical composition of GDY. The presence of $sp$ carbon is also demonstrated by the Raman spectrum (Fig. 2f), in which the characteristic bands at 1932 and 2131 $cm^{-1}$ deriving from the vibration of the conjugated diyne linkage are clearly present[58]. A flat film of GDY is obtained by spin-coating its suspended dispersion in $N,N$-dimethylformamide (DMF) (Supplementary Fig. 1c, 1d), with a thickness of approximately 400 nm (Supplementary Fig. 1e).

To illustrate the dynamics of the migration of anions and cations, ten current–voltage (I–V) sweeps were performed for Li-GAS and Na-GAS (Fig. 2g, h). An obvious negative differential resistance (NDR) phenomenon is observed in Li-GAS (Fig. 2g), which is probably due to the migration of $Li^+$ to the surface and plane of GDY. In this way, an electrochemical doping process occurs. However, two groups of NDRs appear in the I–V curve of Na-GAS, of which the NDR at the low potential (<2 V) is more pronounced (Fig. 2h). The obvious NDR in Na-GAS might be caused by the large internal field formed by the interface ions[59]. In the positive sweep range, $Li^+$ cations are doped in GDY, and anions gradually accumulate at the interface, forming an internal field. The intensity of this internal field will temporarily exceed the applied electric field, manifesting as an NDR phenomenon and a dedoping process. As the scanning window becomes narrower, the NDR phenomenon disappears, manifesting the effect of the interface pseudocapacitance (Supplementary Fig. 2a, d). The first eight I–V sweeps for Li-GAS show good reversibility in the low voltage range (Supplementary Fig. 2e, f). The response of Na-GAS in the low voltage range is repeatable after the first few sweeps (Supplementary Fig. 2g, h). Therefore, positive and negative pulses with different amplitudes could have an effect on the synaptic weight under the action of the interfacial capacitance and electrochemical doping from the analysis of the I–V curves. The current response range of Li-GAS is larger, which is also a reflection of the fact that GDY has a better storage capacity for $Li^+$ ions.

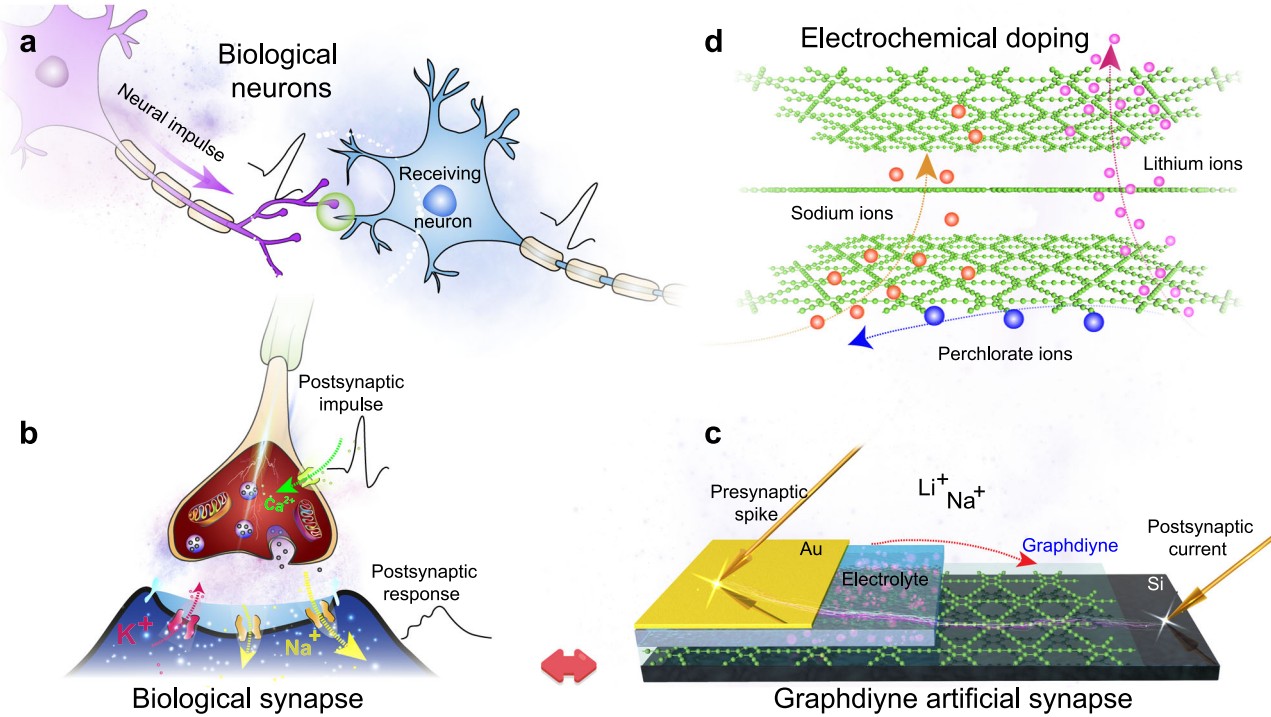

**Fig. 1 Schematic illustration of GASs. a** Signal transmission between neurons. **b** Ion flows in the synaptic cleft. **c** Junction-type GAS. **d** Dynamic diffusion process of ions between GDY layers.

**Short-term plasticity and ultrasensitivity of the GASs.** Synaptic plasticity is modulated by presynaptic action potentials, which can trigger $Ca^{2+}$ influx and release excitatory or inhibitory neurotransmitters to strengthen or weaken information transmission. Presynaptic pulses with different amplitudes, durations, frequencies, and numbers are processed by the synaptic device to output different types of current signals, which is spike-dependent plasticity. In our synaptic devices, Li- and Na-GASs, varying degrees of short-term enhancement are demonstrated. Under the same presynaptic pulse (+2 V, 440 ms), it is easier for $Li^+$ ions to be activated and to migrate in the corresponding synaptic device (Fig. 3a) due to their smaller ion radius and lower diffusion barrier[40,60]. GDY can store more $Li^+$ ions, and thus, Li-GAS exhibits a greater synaptic weight, which is consistent with the charge capacity of GDY regarding Li and Na ion storage[40]. With the gradual increase in the pulse amplitude (from 0.5 to 4 V) for Li-GAS and Na-GAS, the peak value of the postsynaptic current increases stepwise from sub-nA to tens of nanoamps, which is typical spike-voltage-dependent plasticity (Fig. 3a)[61]. This plasticity can also be observed under the action of negative pulses (Supplementary Fig. 3a). Under a single positive or negative pulse stimulus, the surge current decays to the level of ± 0.2 nA in a short time (<6 s), showing obvious bidirectional short-term plasticity of the GASs (Supplementary Fig. 3b, c). Successive pulse stimulation in a short interval ($\Delta t < 3$ s) produces paired current peaks of different heights (A1 and A2). As the time interval increases, the PPF effect (A2/A1×100%) becomes weaker (Fig. 3b), which may result from the rapid diffusion of ions provided by the large pores of the GDY network. At the same pulse interval, Na-GAS exhibits a higher facilitation index than Li-GAS. The facilitation index under negative pulses is relatively low, which might be attributed to the volatile effect required for short-term plasticity.

Negative pulse sequences with different frequencies have an effect on the peak postsynaptic current. The higher the pulse rate is, the more obvious the gain (A10/A1×100%) (Fig. 3c),

demonstrating typical spike-rate-dependent plasticity (SRDP)[62]. This short-term enhanced plasticity enables the GASs to act as a dynamic filter for information transmission[32]. It is obvious that Li-GAS shows better dynamic filtering (Fig. 3d), which is consistent with the previous PPF effect (Supplementary Fig. 4a and 4b). As the number and duration of pulses increase, the current gain becomes less pronounced during the charging process, and the current decay is still fast during the discharging process (Supplementary Fig. 4c–f). The reason for this rapid discharge process could be attributed to ion migration or interface depolarization[30]. Such a rapid volatile process is important for real-time imaging applications through the construction of a synaptic array (Fig. 3e). If the letters G, D, and Y are input into the array (9 × 9), then the 81-pixel image formed by the array can be refreshed in a short time due to the exemplary short-term plasticity of the synaptic unit. When continuous nonidentical negative pulses are applied, the current peaks triggered by pulses of the same amplitude are of equal height, and repeatable short-term plasticity is observed (Fig. 3f). The pulses with different amplitudes correspond well to the different discharge current peaks, which further indicates the good stability and repeatability of the GASs in comparison with solution-processed reduced graphene oxide[63]. Only 0.22 s is required to encode the postsynaptic current triggered by the presynaptic pulses of different amplitudes into a pattern with obvious contrasts (Supplementary Fig. 5a, b). Such a rapid deintercalation of ions should be attributed to the triangular macroporous structure of the GDY network[55,57].

Moreover, when 10 and 15 consecutive negative pulses (−3.5 and −5 V) are applied to Li-GAS, the peak current increases and then quickly declines within a few seconds after removing the pulse (Supplementary Fig. 5c). As the pulse amplitude gradually weakens to the range of 80–20 mV (Fig. 3g), plasticity can still be emulated (Fig. 3g). Such ion diffusion dynamics also occur in Na-GAS under lower operating voltages (≤20 mV), resulting in remarkable pulse sensitivity (Fig. 3g). The average power

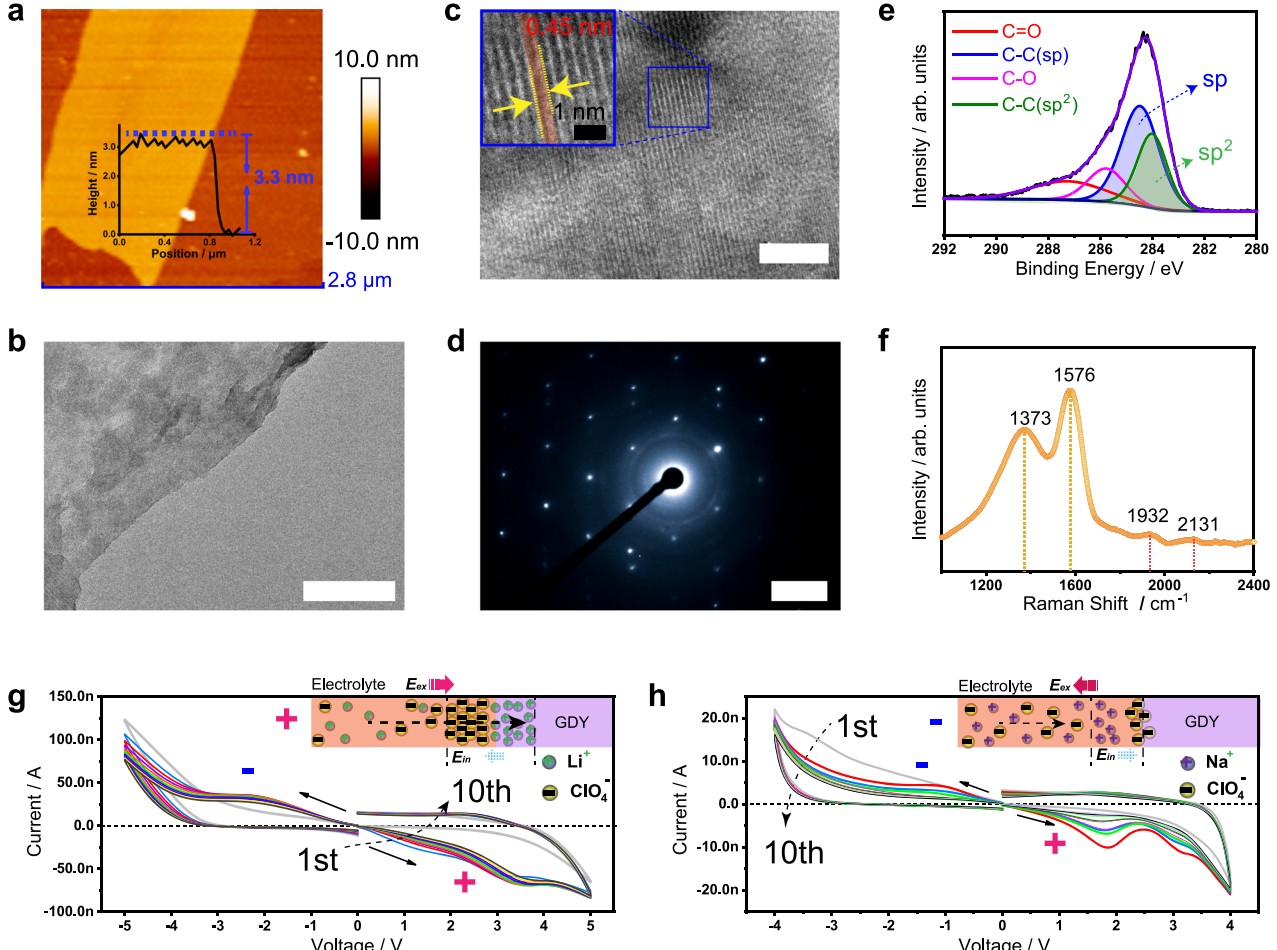

**Fig. 2 Characterizations of GDY and GASs. a** AFM image, **b** TEM image (scale bar: 200 nm), **c** high-resolution TEM image (scale bar: 5 nm), **d** SAED pattern (scale bar: 5 1/nm), **e** high-resolution XPS C 1 s spectra, and **f** Raman spectrum of GDY. **g**, **h** I–V curves measured in sweep cycles of 0 to 5 V and 0 to −5 V for Li-GAS and Na-GAS, respectively. Inset: Schematic illustrations of ion dynamic diffusion under positive and negative pulses.

consumed by a single synaptic event is 16.7 fW (Supplementary Fig. 5d)[3,4,64], which is orders of magnitude lower than the biological level and the most competitive value thus far among the two-terminal devices (Supplementary Table 1). The sensitivity to presynaptic pulses, ultralow power consumption, and significant volatility are integral to constructing a bioinspired ionic sensory/motor system.

**Electronic characteristics and thermal and environmental stability of the GASs.** To better understand the diffusion behavior of alkali metal ions in the GDY network, first-principles calculations are carried out using the Vienna ab initio simulation package (VASP) at the Perdew–Burke–Ernzerhof (PBE)-generalized gradient approximation (GGA) level (see "Methods" for details)[65]. After being fully optimized, the lattice parameters of GDY are $a = b = 9.46$ Å (Supplementary Fig. 6a). The band structure and partial density of states (PDOS) are calculated, demonstrating a bandgap of ~0.50 eV, which is in line with the literature and suggests the intrinsic semiconductor character of GDY (Supplementary Fig. 6b, c). Such results are similar to those reported for arsenicated triphenylene-graphdiyne and single-layer $C_{12}N_2$ (0.47 and 0.50 eV, respectively)[66,67]. When Li or Na ions are adsorbed on the delocalized π-conjugated surface of GDY, two possible adsorption points could be chosen, namely, the center of the benzene ring (A) or above the large triangular pore (B) (Supplementary Fig. 6d)[66–68]. The total energies of Li or Na ion

adsorption on GDY at the A and B sites are calculated to be −154.69 and −155.22 eV for Li and −154.11 and −155.13 eV for Na, respectively. Thus, the B site is the most stable site for the adsorption of both Li and Na and is taken in the following calculations. The optimized structure of Li-adsorbed GDY (Li-GDY) is shown in Fig. 4a. The band structure (Fig. 4b) and PDOS of Li-GDY (Fig. 4c) show no obvious change from those of intrinsic GDY. However, the Fermi level of Li-GDY moves upward to the conduction band, showing an enhancement of the electrical conductivity in comparison with GDY. Additionally, the band structure (Fig. 4e) and PDOS of Na-adsorbed GDY (Na-GDY) (Fig. 4f) show no obvious change from those of GDY, with the Fermi level moving upward to the conduction band.

The calculation of the charge density difference ($\Delta\rho$) of Li-GDY (Supplementary Fig. 6e), defined as $\Delta\rho = \rho_{Li-GDY} - \rho_{Li} - \rho_{GDY}$, in which $\rho_{Li-GDY}$, $\rho_{Li}$, and $\rho_{GDY}$ represent the charge density values of Li-GDY, pristine Li atoms and pristine GDY, respectively, suggests an electron net increase around the GDY structure and an electron net decrease around the Li atom, demonstrating a transfer of electrons from the Li atom to GDY. Similar electron transfer is observed for Na-GDY (Supplementary Fig. 6e). To explore the dynamics of Li and Na ions in their GAS synaptic devices, we calculate the diffusion energy barriers for a single Li or Na ion from the most stable adsorption point in GDY to the most stable adsorption site at the adjacent positions. The diffusion barriers for a single Li or Na atom diffusing at the surface of GDY are calculated to be 0.54 and 0.72 eV for Li-GDY

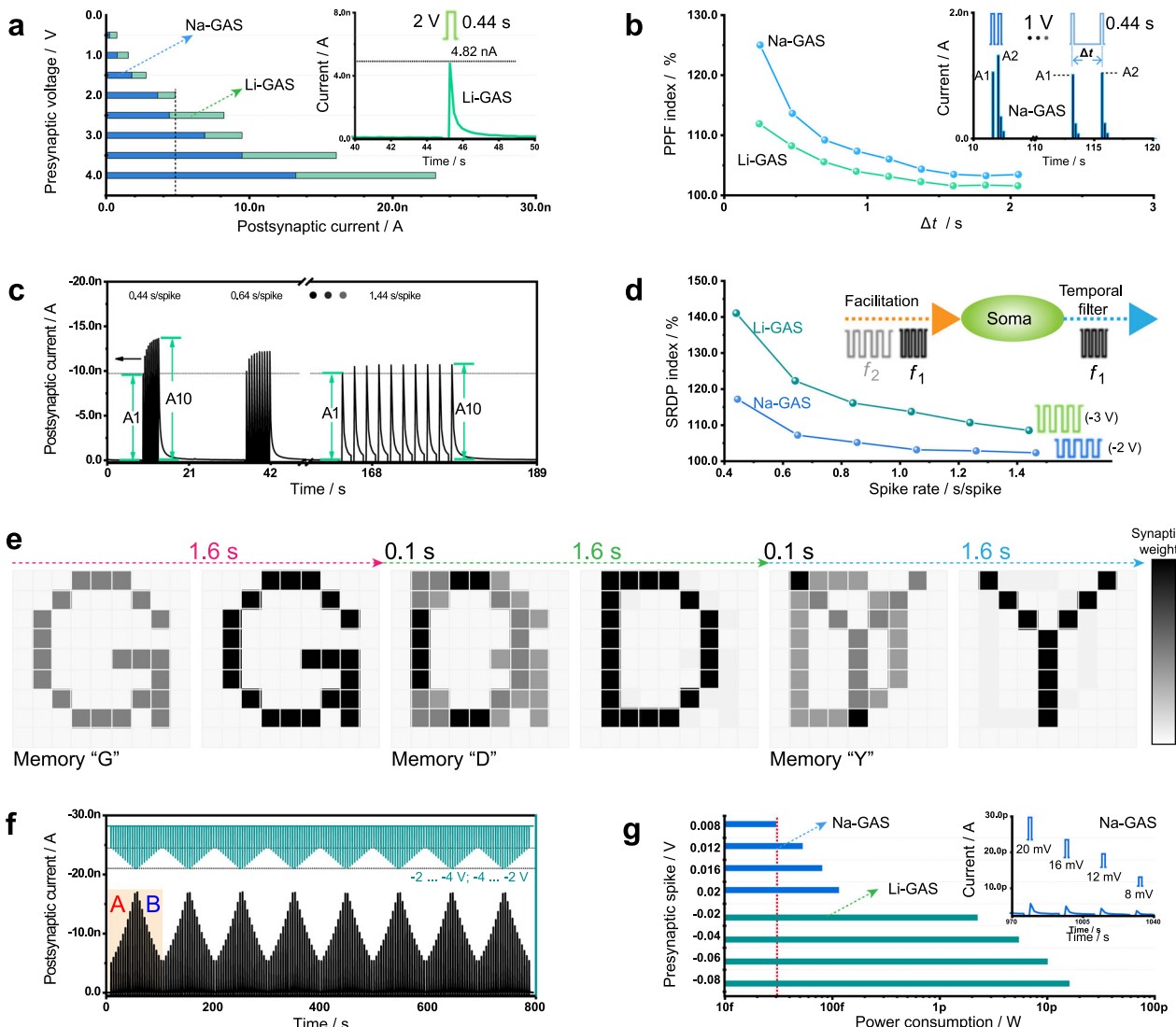

**Fig. 3 Short-term plasticity in Li- and Na-GASs. a** Peak value of the postsynaptic current with different positive pulse amplitudes in Li-GAS and Na-GAS, respectively. Inset: Postsynaptic current triggered by a single spike in Li-GAS. **b** PPF behavior emulated by two consecutive positive pulses (+1 V) in Li-GAS and Na-GAS. Inset: Postsynaptic currents triggered by two spikes in Na-GAS. **c** Postsynaptic current at different frequencies in Li-GAS. **d** Gain of postsynaptic currents (SRDP index; A10/A1×100%) plotted as a function of presynaptic spike rate in Li-GAS (-3 V) and Na-GAS (-2 V). **e** Real-time storage and transformation of letters G, D, and Y. **f** Postsynaptic current triggered by a nonidentical negative pulse sequence in Li-GAS. **g** Impulse responsivity at the millivolt level and corresponding power consumption in Li-GAS and Na-GAS. Inset: Postsynaptic currents triggered by a series of positive presynaptic spikes with voltage amplitudes from 8 to 20 mV in Na-GAS.

and Na-GDY, respectively (Fig. 4g, h). The larger diffusion barrier of Na-GDY than of Li-GDY well explains the experimental fact that Li$^+$ ions migrate easier in the corresponding GAS synaptic device. Interestingly, the reported boron-GDY has a lower Li$^+$ diffusion barrier when Li$^+$ is on top of the boron atom and can accommodate multiple light-metal dopants (Li, Na, K, Ca)[69,70].

The thermal and environmental stability of the fabricated devices is experimentally studied. Temperature and humidity have major influences on the I–V curve displayed by the devices (Fig. 4I, j; Supplementary Fig. 7). As the heat treatment temperature of the device rises to 353 K, an inconspicuous bulge in the I–V curve at ~2.05 V (Fig. 4i, Supplementary Fig. 7a-7c) becomes increasingly acute. The current window displayed by the device does not change much, and the curve is relatively stable. The current window displayed by the device gradually changes as the ambient humidity increases, and the curve is still stable at a humidity up to 35% (Fig. 4j, Supplementary Fig. 7d–7f). Li-GAS

is more sensitive to environmental temperature and humidity (Fig. 4k). After undergoing specific thermal stability (heating temperature: ≤ 353 K) and environmental stability (relative humidity: ≤ 65%) tests, the electrolyte film on the two devices is not severely damaged (Supplementary Fig. 8), demonstrating the reliable stability of the prepared devices.

**Integration, parallel processing, and artificial efferent neuron actuation of the GASs.** Electrolyte-based devices can convert biological ionic/chemical signals into electrical signals, and therefore, attractive dendritic integration can be realized[15]. Multiple input signals applied to the top of the GAS will be integrated and output at the bottom (Fig. 5a). First, the rules of spatiotemporal learning of two synapses are emulated. The two synapses exhibit synaptic connections with different strengths under different amplitude pulse stimulation of Li-GAS and Na-GAS (Fig. 5b). When the two synapses act simultaneously, the

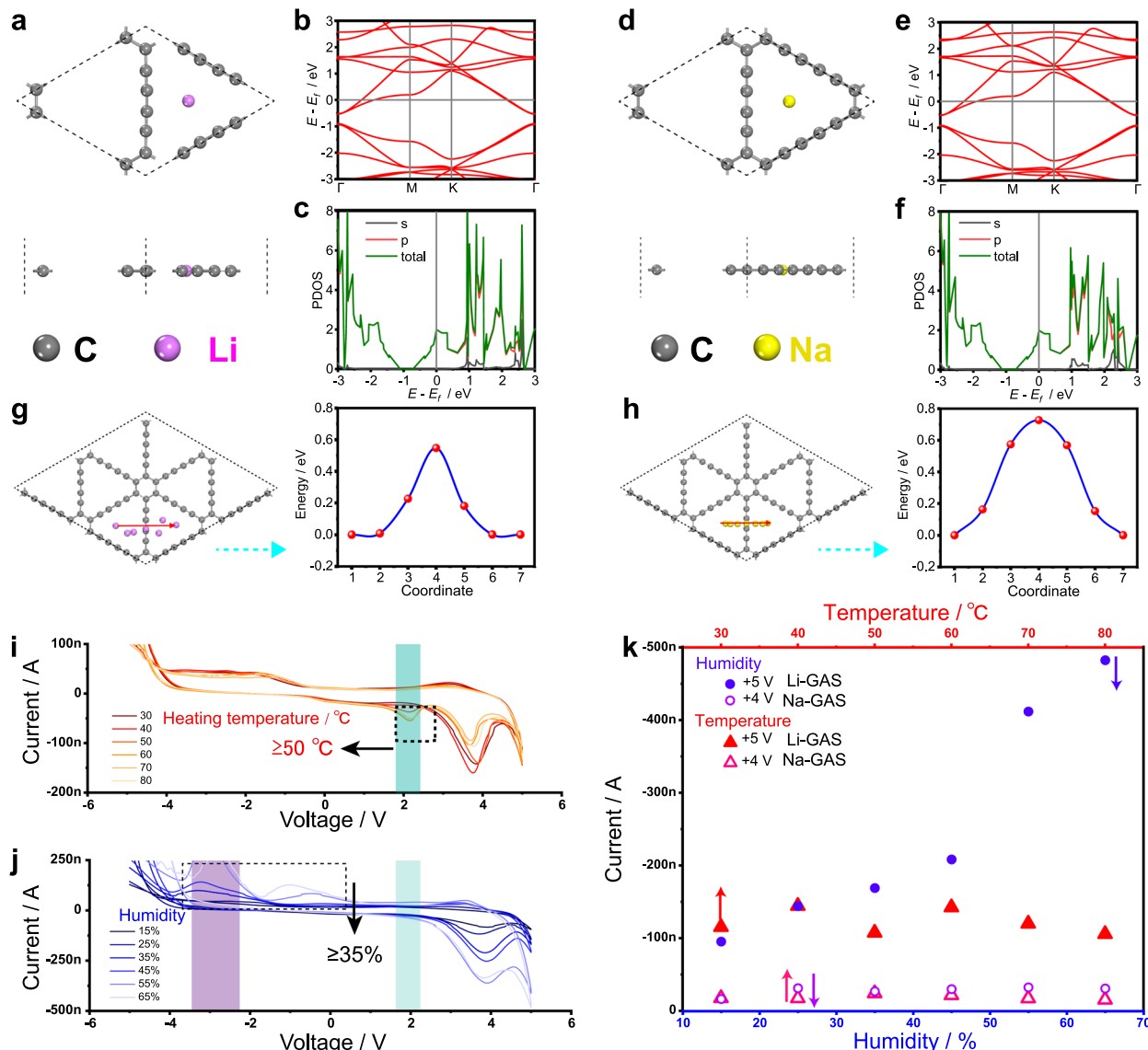

**Fig. 4 Density functional theory calculations and stability of Li- and Na-GASs. a** Top and side views of the optimized structure of Li-GDY. **b** Band structure of Li-GDY. **c** PDOS of Li-GDY. The Fermi level is set to zero. **d** Top and side views of the optimized structure of Na-GDY. **e** Band structure of Na-GDY. **f** PDOS of Na-GDY. The Fermi level is set to zero. Diffusion pathways and corresponding diffusion energy profiles of a single **g** Li or **h** Na ion. The red arrows represent the diffusion directions. I–V curves of Li-GAS in a sweep cycle of −5 to 5 V measured **i** after heat (for 10 min at each temperature from 30 to 80 °C) and **j** humidity (for an hour at each relative humidity from 15% to 65%) treatment. **k** Current value of the devices after treatment in different environments. The current value at a certain voltage (+4 and +5 V for Na-GAS and Li-GAS, respectively) is obtained from the I–V curve for the first cycle of the two devices under different processing conditions.

nonlinear increase in the output is prominent. When one of the two synapses is triggered earlier than the other ($\Delta T < 0$) by less than 4 s, the output postsynaptic current gradually increases to 340% and 250% of the initial output of Li-GAS and Na-GAS (Supplementary Fig. 9a), respectively. Through the input of multiple pre-neurons, interesting temporal features of dendritic connections are observed. This also shows that the synergy of multiple synapses (synapse 1 and synapse 2) can significantly enhance the synaptic weight compared to different numbers of repeated stimulations applied to a single synapse (Fig. 5c). In this way, the gain obtained by parallel processing of multiple synapses is much higher than that obtained with a single synapse under spike-dependent plasticity. Distributing the weight to single synaptic units is the essence of parallel processing. The shared information transmission by multiple synapses better resembles biological information integration than transmission by single synapses.

Similarly, the gain of the postsynaptic current triggered by presynaptic pulses of different durations for a single synapse is still not as significant as the gain obtained by the integration of multiple synapses (Fig. 5d). Therefore, logic operations can be realized according to the different gains of one synapse and two synapses at different durations (Supplementary Fig. 9b). Meanwhile, modulation of the neural responses can be mimicked under the synergy of two synapses. Then, shunting inhibition, a mechanism for regulating the neural response, can be achieved by applying excitatory and inhibitory stimulations (positive and negative pulses) to the two synapses (Fig. 5e). With the gradual increase in a presynaptic inhibitory pulse, the postsynaptic current gradually weakens until it disappears. This suggests that the output can be both magnified and reduced during information integration.

Parallel processing of neuromorphic biological signals is of crucial importance in the novel computational paradigm[15]. It

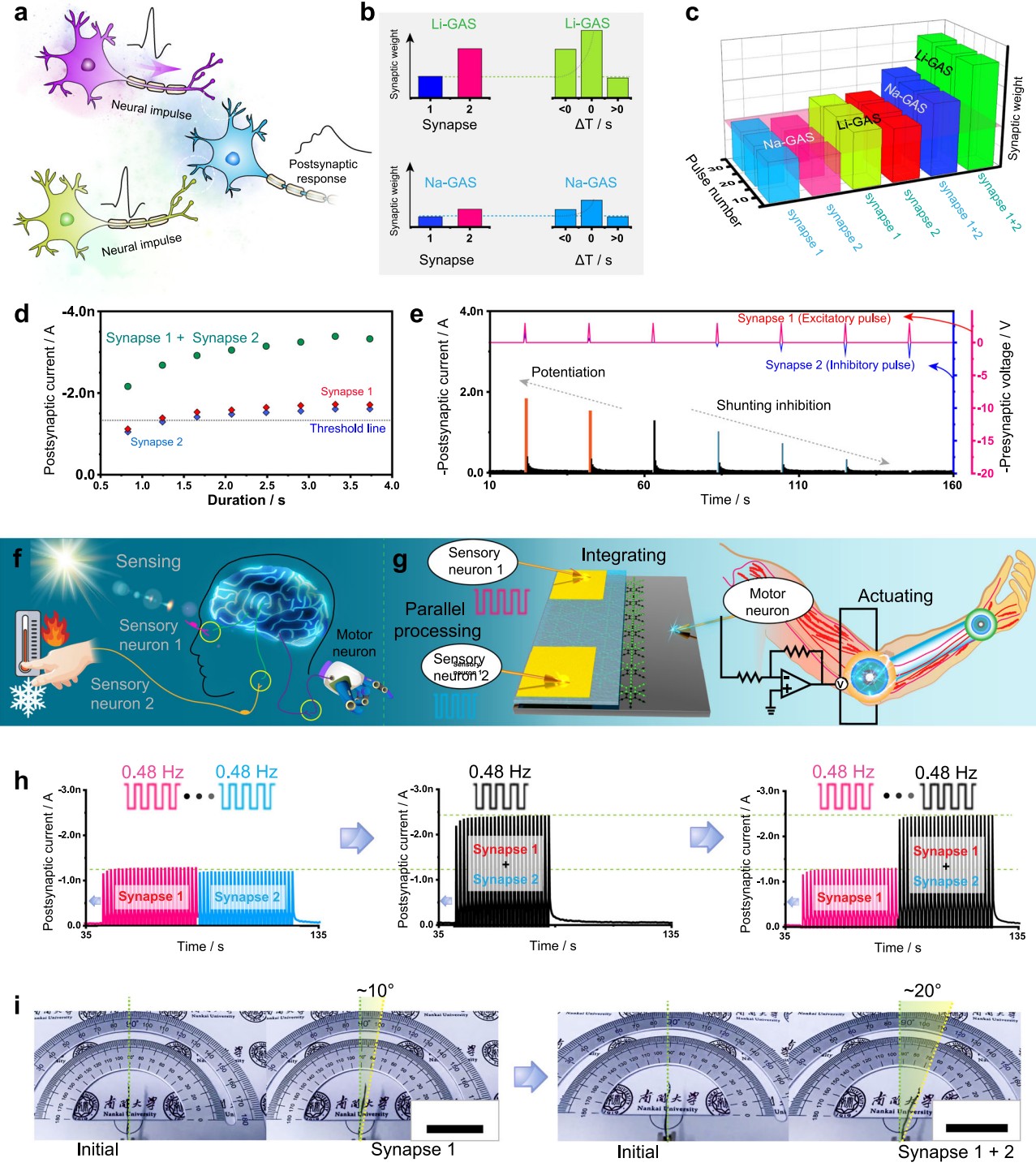

**Fig. 5 Dendritic integration function in Li- and Na-GASs. a** Schematic illustration of spatiotemporal neural networks in the GASs. **b** Synaptic weight (current) triggered by a single pulse and a pair of spatiotemporally correlated pulses in Li-GAS and Na-GAS. **c** Synaptic weight (current peak value) obtained under different numbers of applied pulses ($N = 10$, 20, and 30) in Li-GAS and Na-GAS. **d** Postsynaptic current obtained under different durations of applied spikes in Li-GAS. **e** Postsynaptic currents triggered by a pair of presynaptic (excitatory and inhibitory) pulses for emulating the shunting inhibition function. Schematic illustration of **f** a biological sensorimotor nerve and **g** an artificial efferent nerve, coupling GAS processing elements with artificial muscles, to transduce signals from receptors to motor neurons. **h** Postsynaptic current triggered by two presynaptic inputs at 0.48 Hz from within the same time period to at different time periods. **i** Digital images of the actuator flexion under 0.8 Hz pulse sequences for single and double synapses. The scale bar corresponds to 2 cm.

remains a challenge for synaptic units to perform real-time parallel processing of external perception information (light and temperature) transmitted by multiple sensory neurons and to make artificial efferent nerves respond differently to the

integrated information (Fig. 5f). Here, impulse stimuli of different frequencies are applied to the top electrode of the GAS as perceptual information from the outside, and artificial muscles are connected to the bottom electrode of the GAS through a

circuit to construct artificial efferent neurons (Fig. 5g). Initially, under a single frequency (0.48 Hz) input, the currents generated by two independent synaptic units are similar, and their overall outcome is equal to the peak current of the two synapses in the same time period (Fig. 5h). If the two synapses can simultaneously receive signals provided by peripheral devices in the same time period, then the signal reflected in the post-neuron is the superposition of the signals obtained by the two independent synapses. The effect of integrating multiple inputs and accumulating outputs is significantly higher than that of a single synapse. The results obtained under low-frequency inputs (0.48 and 0.8 Hz; 0.344 and 0.60 Hz) are consistent, which means that the output results of multiple presynaptic pulses in the same time period are accumulated in parallel (Supplementary Fig. 10). Meanwhile, the peak shape of the post-neuronal signal can clearly reflect the time period during which the two synapses receive signals of different frequencies from the peripheral unit, and the frequencies of the presynaptic pulses can be inferred from the different peak-to-valley spacings (Supplementary Fig. 10e). As the pulse frequency increases from 2.34 to 5.60 Hz, the postsynaptic current gradually increases, and good parallel processing capabilities and short-term plasticity can be maintained (Supplementary Fig. 11). Hence, the GAS can identify the frequency of presynaptic pulses to a certain extent to infer and analyze the sensory information transmitted from afferent nerves. Finally, artificial efferent nerves are constructed to implement the integrated output of multiple synaptic inputs (Supplementary Fig. 12). With only one input terminal, the generated output signal reaches the motor neuron via the efferent nerve to trigger artificial muscle bending (Fig. 5i). When two sets of inputs are applied, the signals are processed in parallel and output by the GAS and finally transmitted along the efferent nerve to drive the artificial muscle to produce greater bending. Multiple external inputs are integrated into the GAS, and the cumulative output drives the artificial muscle. Such a GAS element can process in parallel and integrate the received signals in real time, and can be combined with a multifunctional actuator to sense or control objects, and can be coupled with a variety of receivers to sensitively reflect external environmental information[1,12,71].

## Discussion

Inspired by somatosensory nerves, we have designed and demonstrated the first graphdiyne-based artificial synapse components with parallel processing and information integration capabilities. Benefiting from the special working mechanism of ion migration and good electron transport properties, the GAS exhibits essential short-term synaptic behavior, which is conducive to its application in intelligent mechanotransduction systems and biohybrid systems. Furthermore, the GAS exhibits an ultralow voltage response and sub-biological power consumption, which is competitive with other two-terminal resistive switches (e.g., memristors), but a dilemma remains in the large-scale preparation of single- and multilayer GDY. At the same time, the GAS can withstand certain heat and humidity conditions. As an exceptional parallel processing unit, the GAS can identify the frequency (≤5.6 Hz) of presynaptic inputs to a certain extent based on the postsynaptic current to infer and analyze the sensory information transmitted from afferent nerves. By connecting the GAS and artificial motor neurons, artificial efferent nerves are constructed to drive artificial muscles to bend. The GAS can process multiple sets of inputs in parallel and integrate the output to control the degree of bending of artificial muscles. In addition, GDY has outstanding biological activity due to its active acetylene unit, which has already emerged in the fields of biosensing, drug delivery, and living micromotors[46,72,73]. Hence, GDY can be

viably coupled with biological presynaptic neurons due to its biocompatibility to form a bioartificial hybrid system to complete spike transmission and plasticity processing[9]. A functional bio-hybrid GDY system, a bioelectrochemical signal input terminal and a neuromorphic GAS output terminal can be conceived to demonstrate the regulation of synaptic weights in neuromorphic-based prosthetics. Furthermore, it may be more interesting to construct an all-GDY efferent nerve by combining our GAS and GDY-based artificial muscles in the future[74]. Therefore, with the implementation of real-time parallel processing, integration and actuation, the artificial synapse has the potential to be a key processing element in artificial sensorimotor nerves and biohybrid systems of soft electronics, neurorobotics, smart prostheses, and brain–computer interfaces.

## Methods

**Fabrication and characterization of multilayer GDY films**. HEB-TMS (8 mg) was added to 120 mL of degassed dichloromethane and stirred for 10 min. Then, 100 μL of tetrabutylammonium fluoride (1 M in THF, 100 μmol) was injected. The mixture was stirred for another 15 min under an argon atmosphere in the dark. The subsequent reaction could be carried out without purification. The multilayer graphdiyne film was prepared via a liquid/liquid interfacial reaction. Under an argon atmosphere at room temperature, 10 mL of HEB in dichloromethane (0.1 mM) was added to a glass cylinder. Next, 12 mL of deionized water was added to form a layer between the two separate phases. Then, 8 mL of mixture solution consisting of 0.01 M copper acetate and 0.25 M pyridine was dropped slowly into the aqueous phase. The system was kept undisturbed for more than 24 h, and a brown film could be observed at the interface. The reagent was removed from the glass cylinder, and the film could be collected, filtered through a nylon membrane with a 100-nm pore size and washed with HCl (1 M, 10 mL) and pure water (10 mL). The morphology of the as-prepared GDY was characterized by SEM using a QUANTA FEG 450 field-emission microscope. TEM images were obtained using an FEI Talos F200X microscope. The crystal features of GDY were characterized by high-resolution TEM (JEM-2800), and the thickness of GDY and the surface of electrolytes were characterized by AFM (Bruker, DIMENSION ICON). The Raman spectrum of GDY was obtained using a high-resolution confocal Raman microscope (TEO, SR-500I-A) at 532 nm excitation. XPS was conducted with a Thermo Scientific ESCALAB 250Xi instrument.

**Fabrication of Li-GDY and Na-GDY synaptic devices and electrical measurements**. A doped Si substrate was cleaned by sonication in deionized water, acetone and 2-propanol, boiled in 2-propanol, and then treated with ultraviolet ozone. GDY (5 mg) was placed in DMF (10 mL) and dispersed ultrasonically for 10 min. The resulting GDY dispersion (80 μL) was spin-coated at 800 rpm on the treated substrate and annealed at 80 °C for 20 min. Lithium-ion and sodium-ion solid polymer electrolytes (Li-SPE and Na-SPE) were obtained by mixing and stirring PEO powder (0.8 g) with the corresponding perchlorate (0.1 g) in acetonitrile (10 mL). The two electrolytes were spin-coated on the GDY film, and the devices were annealed at 90 °C in a nitrogen-filled glove box for 20 min, after which Au-dot electrodes were subsequently deposited to obtain Li-GAS and Na-GAS. All electrical measurements were performed using a Keithley 4200A semiconductor parameter analyzer in a nitrogen-filled glove box with moisture and oxygen contents of less than 0.1 ppm. To test the thermal and humidity stability, the device was placed in an environment with different temperatures and humidities, and then, the device was placed in a glove box for electrical testing. The device was heated from 30 to 80 °C at an interval of 10 °C, and the time for each heat treatment was 10 min. The relative humidity in the environmental treatment of the device ranged from 15% to 65%, and the time for each humidity treatment was 1 h.

**Fabrication of the electrolyte layer and actuator**. The electrolyte layer was fabricated by dissolving poly(vinylidene fluoride-co-hexafluoropropylene) (PVDF-HFP) and 1-ethyl-3-methylimidazolium tetrafluoroborate (EMIBF$_4$) in 2 mL of DMF at 60 °C for 1 day to obtain a uniform solution. A glass mold was used to prepare the electrolyte layer by solution casting in a N$_2$ atmosphere for 1 day at room temperature, and then, the electrolyte layer was obtained by peeling it from the glass. The electrolyte layer was sandwiched by carbon nanotube (CNT) electrodes, which were pressed at 70 °C for 2 min to fabricate the actuator. Subsequently, the as-prepared actuator was aged under reduced pressure at room temperature for one day and then cut into strips of the same dimensions (20 × 2 mm$^2$) for further measurements.

**Construction of the synaptic device-amplifier circuit-polymer actuator system**. To operate the actuator, we introduced an operational amplifier to output the desired voltage. The bottom electrode of the synaptic device was connected to the amplifier circuit to convert currents to output voltages such that the actuator could be operated. One end of the bottom electrode of the synaptic device was coated

with silver paste and dried in air. Copper wires were used for connection. The amplifier circuit amplified the input voltage by 250,000 times to reach the working voltage of the actuator of ~ 3 V.

**First-principles calculations**. We performed density functional theory (DFT) simulations using the Vienna ab initio simulation package (VASP) with the exchange and correlation functionals of the PBE form and the generalized gradient approximation (GGA). The energy cutoff was 520 eV. For geometry optimization, convergences of $10^{-4}$ eV for energy and 0.02 eV $Å^{-1}$ for force were applied. For electronic calculations, a convergence of $10^{-6}$ eV for energy was applied. The $k$-point mesh was sampled by the Monkhorst–Pack method with a separation of 0.02 $Å^{-1}$. The vacuum space in the vertical direction was set to above 20 Å to avoid periodic images. The climbing image nudged elastic band (CiNEB) method was applied to calculate the ion diffusion. We built a 2 × 2 supercell of GDY (18 C atoms in total) to avoid Li/Na interactions in different supercells.

**Reporting summary**. Further information on research design is available in the Nature Research Reporting Summary linked to this article.

## Data availability

The data that support the findings of this study are available within the paper and its Supplementary Information files. Additional data and files are available from the corresponding author upon reasonable request. Source data are provided with this paper.

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

## Acknowledgements

This research is supported by the Tianjin Science Foundation for Distinguished Young Scholars (19JCJQJC61000), National Natural Science Foundation of China (21773168), Key Area R&D Program of Guangdong Province (2018B030338001), Hundred Young Academic Leaders Program of Nankai University (2122018218), Natural Science Foundation of Tianjin (18JCYBJC16000), 111 Project (B16027 and B18030), International Cooperation Base (2016D01025), and Tianjin International Joint Research and Development Center.

## Author contributions

H.W., R.S., and L.S. contributed equally to this work. W.X., J.X., and H.W. conceived the original concept and designed the experiments. R.S., C.L., and H.W. synthesized and characterized the GDY. H.W. fabricated the devices and performed electrical measurements. W.X. and L.S. designed the systems and prepared the circuits. H. Y., J. G., Z. X., Y. N. contributed to analysis and discussion on the data. H.W., R.S., and L.S. wrote the manuscript with input from all the other authors. All authors discussed the results and commented on the manuscript.

## Competing interests

The authors declare no competing interests.
