## [Peer Review File · Nature Communications]

REVIEWER COMMENTS

Reviewer #1 (Remarks to the Author):

In this work, authors proposed graphdiyne-based artificial synapse (GAS) to mimic the biological signal transmission behaviors with, because of their intrinsic short-term plasticity. The constructed structures are also proven to be desirable for ultra-sensitive and power efficient brain-inspired applications. Graphdiyne carbon based nanosheets have been recently experimentally fabricated and such that this study is timely and the obtained results can be potentially useful and attractive. I found this work as an original contribution in the field, opening highly attractive new application areas for graphdiyne and such that I can recommend the publication of this manuscript provided that the authors address the following comments:

1-More discussion and results are required with respect to the thermal and environmental stability of fabricated devices.

2-More results on the electronic characteristics of the fabricated Li- and Na- GASs structures can be useful. I recommend the authors to include some first-principles DFT results to enhance the physical insight. In this context, please comment on and refer to the theoretical works presented in Carbon 141 (2019), 291-303 and Carbon 137 (2018), 57-67 .

Reviewer #2 (Remarks to the Author):

Overall comments:

Although presented manuscript reports interesting improvements in properties for the graphdiyne-based artificial synapse, from my point of view, the article does not provide what we called GROUND-BREAKING RESEARCH RESULTS. These improvements are more suitable to be published in the specific scientific journals such as Advanced Materials, Advanced Science, etc. I therefore would suggest to reject this manuscript and resubmit it later for the specific journal with further improvements specified in the comments below.

Specific comments:

1. The Introduction section does not convey clearly the importance and the novelty of the topic.
2. The synaptic characteristics of such a thick film (400 nm thickness) are not even in the range of simple oxide materials. It should be indicated that in comparative scheme, the 2D nanostructured film of the present research does not have any superiority to other reported counterparts.
3. Explanations in manuscript are not well orchestrated. It is hard to grasp the detailed evidence for each individual property of the synapse.
4. In the section of Results - General concept: Such a long explanation about the behavior of natural synapses is not necessary. The whole paragraph can be summarized in few lines.
5. What is scientific explanation in the following sentence: Under the presynaptic pulses (+2V, 440 ms), it is easier for Li⁺ ions to be activated and to migrate in the corresponding synaptic device (Fig. 3a).

6. Please explain where is the reference for the following sentence: "such a rapid deintercalation of ions should be attributed to the triangular macroporous structure of the GDY network (Supplementary Figure 5).

7. Editorial: Figure 3g, please correct the grammar in the sentence "Average power consumption"

8. How did you calculate the number of 16.7 fW for power consumption of device?

9. Present the last findings about the power consumption of synaptic devices where the energy consumption and device specifications are compared in a Table (in supplementary section).

Reviewer #3 (Remarks to the Author):

In this paper, the Authors investigate new materials for artificial synapses. They designed Graphdiyne-based artificial synapse for mimicking biological synapses and short-term plasticity.

This paper is well written and clear to read. Results and applications of these new artificial synapses are impressive.

However, the main utility of these artificial synapses are not clear. Is it for neuromorphic systems or for bio-hybrid systems or for both ? Could these artificial synapses be embedded in a neuromorphic system ? And what is the maximum frequency these artificial synapses can work ?

I have several comments:

1) Introduction

In the introduction part, it could be nice to add few sentences to explain in which systems these artificial synapses can be added.

L56 It could improve the state of the art of the paper adding these references on bio-hybrid systems: Buccelli, S. et al. A Neuromorphic Prosthesis to Restore Communication in Neuronal Networks. *iScience* 19, 402–414 (2019).

Keren, H., Partzsch, J., Marom, S. & Mayr, C. G. A Biohybrid Setup for Coupling Biological and Neuromorphic Neural Networks. *Front. Neurosci.* 13, 432 (2019).

Serb, A. et al. Memristive synapses connect brain and silicon spiking neurons. *Sci. Rep.* 10, 1–7 (2020).

Mosbacher, Y. et al. Toward neuroprosthetic real-time communication from in silico to biological neuronal network via patterned optogenetic stimulation. *Sci. Rep.* 10, 7512 (2020).

L60 It could improve the state of the art of the paper adding these references on microfluidic synapses and artificial synapses:

Keene, S. et al. A biohybrid synapse with neurotransmitter-mediated plasticity, *Nature Materials*, 19:969-973 (2020)

Levi, T. et Fujii, T. Microfluidic neurons: a new way in neuromorphic engineering?, *Micromachines*, 7:146 (2016)

2) Results

It is not clear for me if you can use your artificial synapse in two mode: excitatory and inhibitory. I can see the shunting inhibition but could you also provide negative current?

As I explained in my general comment, which will be the best system that you can design using your artificial synapses? Neuromorphic chip? Bio-hybrid system? Both?

It could be interested to used your synapses with the most famous neuromorphic chip like Indiveri group, Spinnaker, Loihi, etc..) or even more biomimetic artificial neurons like in Khoiratee et al. work. Here are some references on these neuromorphic chip.

S. Furber, D. Lester, L. Plana, J. Garside, E. Painkras, S. Temple, A. Brown, Overview of the

SpiNNaker system architecture, IEEE Transactions on Computers, 62:12, 2454-2467, 2012
P. Merolla et al., A million spiking-neuron integrated circuit with a scalable communication network and interface, Science, 345:6197, August 2014

M. Davies et al., Loihi: A neuromorphic manycore processor with on-chip learning. IEEE Micro, 38(1):82-99, jan 2018

G. Indiveri et al., Neuromorphic silicon neuron circuits, Frontiers in Neuroscience, 5:73, 2011

F. Khoyratee et al., Optimized real-time biomimetic neural network on FPGA for bio-hybridization, Frontiers in Neuroscience, 13-377, 2019

Could you describe more in details your experiments using artificial neurons, how do you connect your nA current to the other artificial device?

Is it possible to design one system using the same substrate (artificial neurons and artificial synapses) ?

One other point is not clear. Which level of amplitude you need to get answer from your synapses? In the manuscript you explained several times that your using pulse at volt level (ex: +2V, 440ms). But at the en, you talked about mV pulse. Maybe I missed one explanation, but could you describe which exact level of pulse you need to activate the synapses?

In supplementary Figure 7, you show results using low frequency pulse. Usually in neuromorphic systems the spike activity frequency is higher. How your artificial synapses answer to higher pulse frequency (10-200Hz)?

3) Discussion

You described results using artificial muscles. Is your system biocompatible (as Graphdiyne is biocompatible)? If yes, could you culture some neurons on your synapses and test this bio-hybrid system ? For instance, preneuron or postneuron are biological neurons. Could you add this study of feasibility on the discussion part?

Could you compared our artificial synapses with other artificial synapses like Memristor? Advantages and drawbacks

Point-to-Point Responses to Reviewers' Comments

Reviewer #1 (Remarks to the Author):

In this work, authors proposed graphdiyne-based artificial synapse (GAS) to mimic the biological signal transmission behaviors with, because of their intrinsic short-term plasticity. The constructed structures are also proven to be desirable for ultra-sensitive and power efficient brain-inspired applications. Graphdiyne carbon based nanosheets have been recently experimentally fabricated and such that this study is timely and the obtained results can be potentially useful and attractive. I found this work as an original contribution in the field, opening highly attractive new application areas for graphdiyne and such that I can recommend the publication of this manuscript provided that the authors address the following comments:

RESPONSE:

We would like to sincerely thank the reviewer for the very positive comments, which summarize the key findings of our work. We have improved the manuscript according to your instructive suggestions/comments by adding some discussion and results. In particular, we have added experimental results of the thermal and environmental stability of the devices and the calculated results of the electronic characteristics of the fabricated Li- and Na- GAS structures (new Fig. 4, new Supplementary Figs. 6, 7, 8). We believe that these revisions have greatly improved the quality of the work, and it is now suitable for publication in *Nature Communications*.

Please find below our responses (in blue) to each of your specific comments (in black). Revisions to the original article have been highlighted in red.

1. More discussion and results are required with respect to the thermal and environmental stability of fabricated devices.

RESPONSE:

We thank the referee for the helpful suggestions. We have studied experimentally the thermal and environmental stability of fabricated GAS devices (Fig. 4i, j, k, new Supplementary Figs. 7, 8) in the revised manuscript, and have launched some discussions on this.

The temperature range of the thermal stability test of the two prepared devices (Li- and Na-GAS) is from 30 to 80 °C at an interval of 10 °C. The heat treatment time at each temperature is 10 minutes. As shown in Fig. 4i, 4k, and Supplementary Fig. 7a-7c, the current-voltage curves displayed by the two devices after heating at different temperatures are relatively stable.

The relative humidity of the environmental stability test of the two prepared devices (Li- and Na- GAS) ranges from 15% to 65% at an interval of 10 %. The treatment time under each relative humidity is one hour. As Fig. 4j, 4k, and Supplementary Fig. 7d-7f, when the relative humidity of the environment increases, the current window displayed by the two devices remains stable below the relative humidity of 35%, but gradually expands after the humidity increases further.

After the device has undergone different temperature and humidity treatments, it can be found

that there are no obvious cracks in the electrolyte layer by observing the surface of the electrolyte layer of the outermost layer of the device (Supplementary Fig. 8).

In general, through a series of current-voltage curves of the devices after different treatments, it can be seen that the device is thermally stable at the temperature up to 353 K and environmentally stable in a relative humidity of 35%.

Revised parts:

We have added the relevant information to the revised manuscript and Supplementary Information as below.

Page 2 (“Abstract” part):

“It is also found that GAS is thermally stable (at 353 K) and environmentally stable (in a relative humidity extending 35%).”

Page 4 (“Introduction” part):

“The GASs can retain good ion diffusion dynamics under relatively high temperature (~353 K) and humidity (~35%).”

Page 12 (“Results” part):

“The thermal and environmental stability of the fabricated devices has been experimentally studied. Temperature and humidity have major influences on the I - V curve displayed by the devices (Fig. 4i and 4j; Supplementary Fig. 7). As the heat treatment temperature of the device rises to 353 K, an inconspicuous bulge at ~2.05 V in the I - V curve (Fig. 4i, Supplementary Fig. 7a-7c) becomes more and more acute. The current window displayed by the device does not change much and the curve is relatively stable. The current window displayed by the device gradually changes as the ambient humidity increases and the curve is still stable at a humidity up to 35% (Fig. 4j, Supplementary Fig. 7d-7f). It can be found that Li-GAS is more sensitive to environmental temperature and humidity (Fig. 4k). After the experience of specific thermal stability (heating temperature: $\leq 353\text{K}$) and environmental stability (relative humidity: $\leq 65\%$), the electrolyte film on the two devices has not been severely damaged (Supplementary Fig. 8), demonstrating the reliable stability of the prepared devices.”

Page 13 (“Results” part):

“

Fig. 4 Density functional theory calculations and stability of Li- and Na- GASS. *I-V* curves of Li-GAS in a sweep cycle of -5 to 5 V measured **i** after heat (for ten minutes at each temperature from 30 to 80 °C) and **j** humidity (for an hour at each relative humidity from 15% to 65%) treatment. **k** The current value of the devices after treatment in different environments. The current value at a certain voltage (+4, +5 V for Na-GAS and Li-GAS respectively) is obtained from the *I-V* curve of the first circle of the two devices under different processing conditions.”

Page 18 (“Discussion” part):

“At the same time, GAS can withstand certain heat and humidity.”

Page 20 (“Methods” part):

“In order to test the thermal and humidity stability, the device is placed in an environment with different temperature and humidity, and then the device is placed in a glove box for electrical testing. The device is heated from 30 to 80 °C at an interval of 10 °C, and the time for each heat treatment is 10 minutes. The relative humidity of the environmental treatment of the device is from 15% to 65%, and the time for each humidity treatment is one hour.”

Page 34 (“Supplementary Information” part):

“

Supplementary Fig. 7 I - V curves in the sweep cycle of -4 to 4 V measured at room temperature **a** after Na-GAS heat (for ten minutes at each temperature from 30 to 80 °C) and **d** humidity (for an hour at each relative humidity from 15% to 65%) treatment. The current changes of **b** Li-GAS and **c** Na-GAS under different cycles at 30 and 80 °C respectively. The current changes of **e** Li-GAS and **f** Na-GAS under different cycles at 15% and 65% relative humidity respectively.”

Page 35 (“Supplementary Information” part):

“

Supplementary Fig. 8 The surface morphology of the electrolyte in **a** Li-GAS and **b** Na-GAS in

the initial state and after heating and environmental humidity treatment.”

2. More results on the electronic characteristics of the fabricated Li- and Na- GASs structures can be useful. I recommend the authors to include some first-principles DFT results to enhance the physical insight. In this context, please comment on and refer to the theoretical works presented in Carbon 141 (2019), 291-303 and Carbon 137 (2018), 57-67.

RESPONSE:

We thank the referee for the constructive comments. We agree that the electronic characteristics of the fabricated Li- and Na- GASs are important and should be discussed.

We have carried out first-principles density functional theory calculations on the band structure, partial density of states, and the diffusion energy (Fig. 4a-h, and Supplementary Figs. 6). The results suggest that GDY may have potential in electronic devices like the reported arsenicated triphenylene-Graphdiyne and single-layer $C_{12}N_2$, (0.47 and 0.5 eV, respectively) (refs. 66 and 67).

We have also added the related theoretical works in the reference list, including those presented in Carbon 141 (2019), 291-303 and Carbon 137 (2018), 57-67, as refs. 66 and 67. We have also made a brief explanation and comparison in the revised manuscript and the referenced literature (refs. 66, 67, 68, and 69).

Revised parts:

We have added the relevant information to the revised manuscript and Supplementary Information as below.

Page 11 (“Results” part):

“**Electronic characteristics, thermal and environmental stability of GAS.** To better understand the diffusion behavior of alkali metal ions in the GDY network, first-principle calculations have been carried out using the Vienna *ab initio* simulation package (VASP) at the Perdew-Burke-Ernzerhof (PBE)-generalized gradient approximation (GGA) level (see Methods for details). After being fully optimized, the lattice parameters of GDY are $a = b = 9.46 \text{ \AA}$ (Supplementary Fig. 6a). The band structure and partial density of states (PDOS) are calculated, demonstrating a bandgap of $\sim 0.50 \text{ eV}$, which is in line with the literatures and suggests the intrinsic semiconductor characteristics of GDY (Supplementary Fig. 6b and 6c). Such results are in consistence with the reported arsenicated triphenylene-graphdiyne and single-layer $C_{12}N_2$, (0.47 and 0.50 eV, respectively). When the Li or Na ions are adsorbed on the delocalized π -conjugated surface of GDY, two possible adsorption points could be chosen, namely the center of the benzene ring (A) or above the large triangular pore (B) (Supplementary Fig. 6d). The total energies of the Li or Na ion adsorbed GDY at A and B sites were calculated to be -154.69 and -155.22 eV for Li, respectively, and are -154.11 and -155.13 eV for Na, respectively. Thus, B site is the most stable site for the adsorption of both Li and Na, and is taken in the following calculations. The optimized structure of Li adsorbed GDY (Li-GDY) is shown in Fig. 4a. The band structure (Fig. 4b) and PDOS of Li-GDY (Fig. 4c) show no obvious change from those of intrinsic GDY. However, the

Fermi level of Li-GDY moves upward to the conduction band, showing an enhancement of the electrical conductivity in comparison with GDY. Also, the band structure (Fig. 4e) and PDOS of Na adsorbed GDY (Na-GDY) (Fig. 4f) show no obvious change from GDY, with the Fermi level moves upward to the conduction band.

The calculation on the charge density difference ($\Delta\rho$) of Li-GDY (Supplementary Fig. 6e), defined as $\Delta\rho = \rho_{\text{Li-GDY}} - \rho_{\text{Li}} - \rho_{\text{GDY}}$, in which $\rho_{\text{Li-GDY}}$, ρ_{Li} , and ρ_{GDY} represent the charge density values of the Li-GDY, pristine Li atoms, and pristine GDY, respectively, suggests an electron net increase around the GDY structure and an electron net decrease around the Li atom, and demonstrates a transfer of electrons from the Li atom to GDY. Similar electron transfer has been observed from Na-GDY (Supplementary Fig. 6e). To explore the dynamics of Li and Na ions in their GAS synaptic devices, we have calculated the diffusion energy barriers of single Li or Na ions starting from the most stable adsorption point in GDY to the most stable adsorption site at the adjacent position. The diffusion barriers of a single Li or Na atom diffusing at the surface of GDY have been calculated to be 0.54 and 0.72 eV for Li-GDY and Na-GDY, respectively (Fig. 4g and 4h). The larger diffusion barrier of Na-GDY than that of Li-GDY well explains the experimental fact that Li^+ ions migrate easier in the corresponding GAS synaptic device. Interestingly, the reported boron-GDY has a lower Li^+ diffusion barrier when Li^+ is on top of the boron atom and can accommodate multiple light-metal dopants (Li, Na, K, Ca).”

Page 13 (“Results” part):

“

Fig. 4 Density functional theory calculations and stability of Li- and Na- GASSs. **a** Top- and side- views of optimized structure of Li-GDY. **b** Band structures of Li-GDY. **c** PDOS of Li-GDY. The Fermi level is set to zero. **d** Top- and side- views of optimized structure of Na-GDY. **e** Band structures of Na-GDY. **f** PDOS of Na-GDY. The Fermi level is set to zero. The diffusion pathways

and corresponding diffusion energy profiles of a single **g** Li or **h** Na ion. The red arrows represent the diffusion directions.”

Page 21 (“Methods” part):

“**First-principles calculations.** We performed the density functional theory (DFT) simulations using the Vienna *ab initio* simulation package (VASP) with the exchange and correlation functions of the Perdew-Burke-Ernzerhof (PBE) form and the generalized gradient approximation (GGA). The energy cutoff was 520 eV. For geometry optimization, the convergences of 10^{-4} eV for energy and 0.02 eV \AA^{-1} for force were applied. For electronic calculations, the convergence of 10^{-6} eV for energy was applied. The *k*-point mesh was sampled by the Monkhorst–Pack method with a separation of 0.02 \AA^{-1} . Vacuum space is set above 20 \AA in the vertical direction to avoid periodic images. The climbing image nudged elastic band (CiNEB) method was performed to calculate the ion diffusions. We built a 2×2 supercell of the GDY (18 C atoms in total) to avoid the Li/Na interaction in different supercells.”

Page 24 (“References” part):

- “ 65. Kresse, G. & Furthmüller, J. Efficient iterative schemes for *ab initio* total-energy calculations using a plane-wave basis set. *Phys. Rev. B* **54**, 11169-11186 (1996).
66. Mortazavi, B. *et al.* N-, P-, As-triphenylene-graphdiyne: Strong and stable 2D semiconductors with outstanding capacities as anodes for Li-ion batteries. *Carbon* **141**, 291-303 (2019).
67. Mortazavi, B., Makaremi, M., Shahrokhi, M., Fan, Z. & Rabczuk, T. N-graphdiyne two-dimensional nanomaterials: Semiconductors with low thermal conductivity and high stretchability. *Carbon* **137**, 57-67 (2018).
68. Li, X. & Li, S. Investigations of electronic and nonlinear optical properties of single alkali metal adsorbed graphene, graphyne and graphdiyne systems by first-principles calculations. *J. Mater. Chem. C* **7**, 1630-1640 (2019).
69. Mortazavi, B., Shahrokhi, M., Zhuang, X. & Rabczuk, T. Boron–graphdiyne: a superstretchable semiconductor with low thermal conductivity and ultrahigh capacity for Li, Na and Ca ion storage. *J. Mater. Chem. A* **6**, 11022-11036 (2018).
70. Hussain, T. *et al.* Enhancement in hydrogen storage capacities of light metal functionalized Boron–Graphdiyne nanosheets. *Carbon* **147**, 199-205 (2019).”

Page 33 (“Supplementary Information” part):

“

Supplementary Fig. 6 a Top- and side- viewed chemical structures of monolayer GDY. b Band structures of monolayer GDY. c PDOS of monolayer GDY. The Fermi level is set to zero. d Schematic diagram of two possible locations for one Li or one Na atoms. e Top- and side- viewed figures of difference charge density for Li and Na adsorbed GDY. Red and green colors indicate electron net increase and decrease, respectively.”

Reviewer #2 (Remarks to the Author):

Overall comments:

Although presented manuscript reports interesting improvements in properties for the graphdiyne-based artificial synapse, from my point of view, the article does not provide what we called GROUND-BRAKING RESEARCH RESULTS. These improvements are more suitable to be published in the specific scientific journals such as Advanced Materials, Advanced Science, etc. I therefore would suggest to reject this manuscript and resubmit it later for the specific journal with further improvements specified in the comments below.

RESPONSE:

We would like to sincerely thank the reviewer for your helpful comments on our research. Our work applies Graphdiyne, a new type of photoelectron biocompatible material, to neuromorphic signal transduction systems and has great potential in biological hybrid systems. The prepared Graphdiyne artificial synapse has outstanding performance in impulse response, power consumption, short-term plasticity, and parallelism. The construction of an artificial efferent nerve based on GDY here represents the first attempt of a 2D nanostructured film on artificial motor neurons.

In particular, we have tried our best to improve the quality of the manuscript by addressing all the comments from the referees and the editor, and hope that the work is suitable for publication in *Nature Communications*.

Please find below our responses (in blue) to each of your specific comments (in black). Revisions to the original article are indicated in red.

Specific comments:

1. The Introduction section does not convey clearly the importance and the novelty of the topic.

RESPONSE:

Thank you for your helpful comment. According to the work we have done, we have modified the “Introduction” part to clarify importance and the novelty of our work.

Revised parts:

We have revised and added the relevant information to the revised manuscript as below.

Page 3 (“Introduction” part):

“These intriguing **ion shuttle** characteristics inspire a new perspective of constructing GDY-based artificial synapses (GASs) for mimicking synaptic cleft information transmission, **which is promising for plasticity-mediated signal processing and transmission in bio-hybrid systems and artificial sensorimotor systems.**

Inspired by biological motor neuron, an integrated ionic artificial efferent nerve can be constructed to mimic the real-time processing and manipulation of signals. Here, for the first time, a junction-type GAS has been proposed by coupling the GDY film with the solid-state electrolytes for emulating multiple short-term plasticity such as postsynaptic current, paired-pulse facilitation (PPF), and dynamic filtering, with outstanding pulse responsiveness and femtowatt-level energy consumptions. **The GASs can retain good ion diffusion dynamics under relatively high temperature (~353 K) and humidity (~35%). Attempts to exploit the short-term plasticity of GDY in bioinspired analogous efferent nerve** have demonstrated the real-time information integration, the parallel processing capabilities, **and the signal transduction and actuation**, and therefore paved the way to future bioinspired ionotronic **sensorimotor systems.”**

2. The synaptic characteristics of such a thick film (400 nm thickness) are not even in the range of simple oxide materials. It should be indicated that in comparative scheme, the 2D nanostructured film of the present research does not have any superiority to other reported counterparts.

RESPONSE:

Thank you for your helpful comment. Compared with previously reported oxide and 2D material-based artificial synapses, the superiority and novelty of our work can be summarized as follows:

1. With respect to the impulse response and energy consumption exhibited by the device, the performance of our device is not inferior to the reported counterparts. The comparison results between our device and previously reported counterparts can be found in the table (Supplementary Table 1). The relevant information to the previously reported oxide and 2D material-based artificial synapse is highlighted in yellow and green in the revised part, respectively.

2. With respect to the characteristics of the device, ions can shuttle and store on the surface and interlayers of GDY due to their own triangular pores. Because of this, GAS synapses can exhibit good short-term plasticity under the action of positive and negative pulses, competitive with reported counterparts.

3. With respect to the signal transmission function of the device in artificial sensorimotor system, the ion shuttle characteristic of GDY is applied to an integrated ionic artificial efferent nerve to transduce presynaptic signals and to actuate artificial muscles. Because of this, we constructed and demonstrated the first 2D nanostructured film based artificial efferent nerve based on GDY, which is an important additional to the existing bioinspired analogous efferent nerve (R1-R3).

Finally, we have elaborated on the advantages of our devices and cited some necessary references accordingly in the revised manuscript.

References:

- R1. Lee, Y. *et al.* Stretchable organic optoelectronic sensorimotor synapse. *Sci. Adv.* **4**, eaat7387 (2018).
- R2. Shim, H. *et al.* Stretchable elastic synaptic transistors for neurologically integrated soft engineering systems. *Sci. Adv.* **5**, eaax4961 (2019).
- R3. He, K. *et al.* An Artificial Somatic Reflex Arc. *Adv.Mater.* **32**, 1905399 (2019).

Revised parts:

We have added the relevant information to the “Supplementary Information” as below.

Page 32 (“Supplementary Information” part):

“**Supplementary Table 1.** Comparison of pulse amplitude, power and energy consumption of artificial synapse devices.

Device structures	Pulse amplitude / mV	Power consumption / pW	Energy consumption / pJ
Ag-Cluster-Doped TiO ₂ Ref. S1 Memristor size: $\pi \times 50 \times 50 \mu\text{m}^2$	± 1000		SET: 26.0 RESET: 22.9
IGZO/Alkylated Graphene Oxide Ref. S2 Channel length and width: $10 \times 10 \mu\text{m}^2$	-500		Capacitive: 136 Resistive: 14.3

CH ₃ NH ₃ PbBr ₃ Single Crystalline	-30	0.0157	0.0143
Ref. S3			
Channel length and width: 100 × 260 μm ²			
MoS ₂ /DEME-TFSI	2000		Ionotronic: 4.8
Ref. S4	50000		Electronic: 13000
Channel length and width: 9 × 20 μm ²			
WO ₃ /DEME-TFSI	V _G : 600	519	36
Ref. S5	V _{SD} : 300		
Channel size: 500 × 50 μm ²			
PbS Quantum Dots/Ga ₂ O ₃	SET: 120~260	SET: ~1000	/
Ref. S6	RESET: -50~-190	RESET: ~1000000	
Memristor size: π × 50 × 50 μm ²			
FAPbBr ₃	/	/	2300
Ref. S7			
Device area: 0.1 mm ²			
Silicon Nanocrystals	/	/	0.7
Ref. S8	(Optical Pulse)		
Device area: 2 × 2 mm ²			
In-Doped TiO ₂	/	/	2.41
Ref. S9	(Optical Pulse)		
100 μm gap (source-drain)			
IZO/Nanogranular SiO ₂	Spike amplitude:	/	15
Ref. S10	300		
Channel thickness: 20 nm	(Gate bias: -700)		
PEDOT: PSS/Nafion/	/	/	~10
PEDOT: PSS-PEI			
Ref. S11			
Device area: 10 ³ μm ²			
WO ₃ /Nafion-117 resin	250	125000	625
Ref. S12			
Device area: 0.6 × 1.2 mm ²			
Channel length and width: 100 × 500 μm ²			
P3HT: PEO/PS-PMMA-PS/	Presynaptic spike: /		0.00123
EMMI-TFSI	-1		
Ref. S13	(V _D : 20)		
Channel length: 300 nm			
WSe ₂ /PEO-LiClO ₄	100		0.03
Ref. S14			

Source-drain distance: $\approx 1 \mu\text{m}$			
PETE-S/ETE-S: NaCl	$V_G: 0.5$	/	1.1
Ref. S15	$V_G: 20$		1491
Channel length and width: $30 \times 1000 \mu\text{m}^2$			
NaSbS ₂	Constant	bias: /	5.75
Ref. S16	1000		
/			
MXene/PEO: LiClO ₄	80	0.4	-5.6
Ref. S17			
GDY/PEO: NaClO ₄ (this work)	5	0.0167	/
Device area: $\pi/4 \times 0.33 \times 0.33 \text{ mm}^2$			

”

3. Explanations in manuscript are not well orchestrated. It is hard to grasp the detailed evidence for each individual property of the synapse.

RESPONSE:

Thanks for the comment. We have added some necessary explanations to further elaborate the detailed evidence for each individual property of the synapse in the revised manuscript.

Revised parts:

We added and revised the relevant information to the revised manuscript as below.

Page 6 (“Results” part):

“Therefore, positive and negative pulses with different amplitudes could have an effect on the synaptic weight under the action of interfacial capacitance and electrochemical doping from the analysis of the I - V curves. The current response range of Li-GAS is larger, which is also a reflection that GDY has a better storage of Li⁺ ions.”

Page 7 (“Results” part):

“Under the same presynaptic pulse (+2 V, 440 ms), it is easier for Li⁺ ions to be activated and to migrate in the corresponding synaptic device (Fig. 3a) due to its small ion radius and lower diffusion barrier. GDY can store more Li⁺ ions and thus Li-GAS exhibits a greater synaptic weight, which is consistent with the charge capacity of GDY for Li- and Na- ion storages.”

Page 9 (“Results” part):

“When continuous nonidentical negative pulses are applied, the current peaks triggered by pulses of the same amplitude are of equal height and repeatable short-term plasticity is observed (Fig. 3f). The pulses with different amplitudes correspond well to the different discharge current peaks,

which further indicates the good stability and repeatability of GAS, in comparison with solution-processed reduced graphene oxide. It takes only 0.22 s to encode the postsynaptic current triggered by the presynapse of different amplitudes into a pattern with obvious contrasts (Supplementary Fig. 5a and 5b). **Such a rapid deintercalation of ions should be attributed to the triangular macroporous structure of the GDY network.**

Moreover, when 10 and 15 consecutive negative pulses (-3.5 and -5 V) were applied in Li-GAS, the peak current was increased and then quickly declined within a few seconds after removing the pulse (Supplementary Fig. 5c). As the pulse amplitude gradually weakens to the range of **80 to 20 mV (Fig. 3g)**, the plasticity can be still **emulated (Fig. 3g)**. Such an ion diffusion dynamics also occurs in Na-GAS **under lower operating voltage (≤ 20 mV)**, resulting in a remarkable pulse sensitivity (Fig. 3g). The **average** power consumed by a single synaptic event is **16.7 fW (Supplementary Fig. 5d)**, which is orders of magnitude lower than the biological level and the **competitive** value so far in the two-terminal devices (**Supplementary Table 1**). The sensitivity to presynaptic pulses, ultra-low power consumption, and significant volatility is integral to construct a bioinspired ionic sensory/motor system.”

4. In the section of Results-General concept: Such a long explanation about the behavior of natural synapses is not necessary. The whole paragraph can be summarized in few lines.

RESPONSE:

Thanks for the comment. We have modified the “General concept” of “Results” part to streamline these statements about the behavior of natural synapses.

We have revised the relevant information in the revised manuscript as below.

Page 4 (“Results” part):

“Communication between cells hinge on the propagation of action potentials on axons (Fig. 1a) and **the sodium ions (Na^+) eventually flow into the postsynaptic membrane with the flow of calcium ions (Ca^{2+}) when the action potential reaches the front of the junction (Fig. 1b).**”

5. What is scientific explanation in the following sentence: Under the presynaptic pulses (+2V, 440 ms), it is easier for Li^+ ions to be activated and to migrate in the corresponding synaptic device (Fig. 3a).

RESPONSE:

Thanks for pointing out. Yes, the expression is a bit confusing. We have modified this and cited some references in the revised manuscript.

In general, most electrode materials of LIBs do not have sufficiently big interstitial space within their bulk materials to host and transport Na ions in terms of larger Na ion than Li ion. In our case, the structure of GDY provides favorable path and sites to satisfy the diffusion and insertion/extraction of large diameter ions. Strong repulsion among Na atoms and the large diameter make the substantial storage of Na in hexagonal pore difficult (Ref. 40). This

phenomenon is also observed in other carbon materials with many micropores (Ref. 60). GDY can store more Li⁺ ions relatively and thus Li-GAS exhibit greater synaptic weight, which is consistent with the charge capacity of GDY for two ions storage.

Revised parts:

We have revised and added the relevant information to the revised manuscript as below.

Page 7 (“Results” part):

“In our synaptic devices, Li- and Na- GASs, varying degrees of short-term enhancement have been demonstrated. Under the same presynaptic pulse (+2 V, 440 ms), it is easier for Li⁺ ions to be activated and to migrate in the corresponding synaptic device (Fig. 3a) **due to its small ion radius and lower diffusion barrier.**”

Page 23 (“References” part):

- “40. He, J. *et al.* Hydrogen substituted graphdiyne as carbon-rich flexible electrode for lithium and sodium ion batteries. *Nat. Commun.* **8**, 1172 (2017).
60. Yu, Z.-L. *et al.* Ion-Catalyzed Synthesis of Microporous Hard Carbon Embedded with Expanded Nanographite for Enhanced Lithium/Sodium Storage. *J. Am. Chem. Soc.* **138**, 14915-14922 (2016).”

6. Please explain where is the reference for the following sentence: “such a rapid deintercalation of ions should be attributed to the triangular macroporous structure of the GDY network (Supplementary Figure 5).”

RESPONSE:

We have cited relevant references for this sentence in the revised manuscript.

The unique structure of GDY, featuring a large number of nanoscales triangular pores, endows GDY with a great number of Li storage sites and favors the absorption/desorption and diffusion of Li ions, both in-plane and out-of-plane. Therefore, two-dimensional (2D) materials with a larger interlayer space and/or larger pores in the 2D sheets themselves, are required for intercalation and diffusion of Na throughout the electrode material.

Revised parts:

On Page 23 (“References” part), the following references are added:

- “55. Huang, C. *et al.* Graphdiyne for high capacity and long-life lithium storage. *Nano Energy* **11**, 481-489 (2015).
57. Farokh Niaei, A. H., Hussain, T., Hankel, M. & Searles, D. J. Sodium-intercalated bulk graphdiyne as an anode material for rechargeable batteries. *J. Power Sources* **343**, 354-363 (2017).”

7. Editorial: Figure 3g, please correct the grammar in the sentence “Anverage power consumption”

RESPONSE:

We are sorry for the grammar mistake. We have checked through the manuscript and corrected the typos and grammar mistakes.

Revised parts:

We have revised the relevant information to the revised manuscript as below.

Page 9 (“Results” part):

“The **average** power consumed by a single synaptic event is 16.7 fW (**Supplementary Fig. 5d**), **which is** orders of magnitude lower than the biological level and the **competitive** value so far in the two-terminal devices (**Supplementary Table 1**).”

Page 21 (“References” part):

- “3. van de Burgt, Y. *et al.* A non-volatile organic electrochemical device as a low-voltage artificial synapse for neuromorphic computing. *Nat. Mater.* **16**, 414-418 (2017).
4. Karbalaei Akbari, M. & Zhuiykov, S. A bioinspired optoelectronically engineered artificial neurorobotics device with sensorimotor functionalities. *Nat. Commun.* **10**, 3873 (2019).
64. Harikesh, P. C. *et al.* Cubic NaSbS₂ as an Ionic–Electronic Coupled Semiconductor for Switchable Photovoltaic and Neuromorphic Device Applications. *Adv. Mater.* **32**, 1906976 (2020).”

8. How did you calculate the number of 16.7 fW for power consumption of device?

RESPONSE:

Thanks for your helpful comments. The average power consumed by a single synaptic event is 16.7 fW (given by $I_{\text{peak}} \times V$). According to Supplementary Fig. 5d, the average value of the continuously enhanced current peak I_{peak} is 3.34 pA (-16.7/5 pA). Here, the sum of the first five current peaks is 16.7 pA. Finally, we have described the calculation method of power consumption in Supplementary Fig. 5 in the “Supplementary Information” of the revised manuscript.

Revised parts:

We revised the relevant information to the revised manuscript and Supplementary Information as below.

Page 9 (“Results” part):

“The **average** power consumed by a single synaptic event is 16.7 fW (**Supplementary Fig. 5d**), **which is** orders of magnitude lower than the biological level and the **competitive** value so far in the two-terminal devices (**Supplementary Table 1**).”

Page 21 (“References” part):

- “3. van de Burgt, Y. *et al.* A non-volatile organic electrochemical device as a low-voltage artificial synapse for neuromorphic computing. *Nat. Mater.* **16**, 414-418 (2017).

4. Karbalaee Akbari, M. & Zhuiykov, S. A bioinspired optoelectronically engineered artificial neurobotics device with sensorimotor functionalities. *Nat. Commun.* **10**, 3873 (2019).
64. Harikesh, P. C. *et al.* Cubic NaSbS₂ as an Ionic–Electronic Coupled Semiconductor for Switchable Photovoltaic and Neuromorphic Device Applications. *Adv. Mater.* **32**, 1906976 (2020).”

Page 31 (“Supplementary Information” part):

“

Supplementary Fig. 5 d Postsynaptic current triggered by 5 negative and 5 positive pulses with amplitude of $\pm 5 \text{ mV}$ in Na-GAS. Here, the sum of the first five current peaks is 16.7 pA . The average value of the continuously enhanced current peak I_{peak} is 3.34 pA ($-16.7/5 \text{ pA}$). Hence, the average power consumed by a single synaptic event is 16.7 fW (given by $I_{\text{peak}} \times V$).”

9. Present the last findings about the power consumption of synaptic devices where the energy consumption and device specifications are compared in a Table (in supplementary section).

RESPONSE:

Thanks for the valuable suggestion. In the revised Supplementary Information of the manuscript, we have added a table (new Supplementary Table 1) to compare the pulse amplitude, power and energy consumption, and device specifications of the last findings about the power/energy consumption of synaptic devices.

Revised parts:

We have added the relevant information to the “Supplementary Information” as below.

Page 32 (“Supplementary Information” part):

“**Supplementary Table 1.** Comparison of pulse amplitude, power and energy consumption of artificial synapse devices.

Device structures	Pulse amplitude	Power consumption	Energy consumption
	/ mV	/ pW	/ pJ

Ag-Cluster-Doped TiO ₂	±1000	/	SET: 26.0
Ref. S1			RESET: 22.9
Memristor size: $\pi \times 50 \times 50 \mu\text{m}^2$			
IGZO/Alkylated Graphene Oxide	-500	/	Capacitive: 136
Ref. S2			Resistive: 14.3
Channel length and width: $10 \times 10 \mu\text{m}^2$			
CH ₃ NH ₃ PbBr ₃ Single Crystalline	-30	0.0157	0.0143
Ref. S3			
Channel length and width: $100 \times 260 \mu\text{m}^2$			
MoS ₂ /DEME-TFSI	2000	/	Ionotronic: 4.8
Ref. S4	50000		Electronic: 13000
Channel length and width: $9 \times 20 \mu\text{m}^2$			
WO ₃ /DEME-TFSI	V _G : 600	519	36
Ref. S5	V _{SD} : 300		
Channel size: $500 \times 50 \mu\text{m}^2$			
PbS Quantum Dots/Ga ₂ O ₃	SET: 120~260	SET: ~1000	/
Ref. S6	RESET: -50~-190	RESET: ~1000000	
Memristor size: $\pi \times 50 \times 50 \mu\text{m}^2$			
FAPbBr ₃	/	/	2300
Ref. S7			
Device area: 0.1 mm^2			
Silicon Nanocrystals	/	/	0.7
Ref. S8	(Optical Pulse)		
Device area: $2 \times 2 \text{ mm}^2$			
In-Doped TiO ₂	/	/	2.41
Ref. S9	(Optical Pulse)		
100 μm gap (source-drain)			
IZO/Nanogranular SiO ₂	Spike amplitude: /		15
Ref. S10	300		
Channel thickness: 20 nm (Gate bias: -700)			
PEDOT: PSS/Nafion/	/	/	~10
PEDOT: PSS-PEI			
Ref. S11			
Device area: $10^3 \mu\text{m}^2$			
WO ₃ /Nafion-117 resin	250	125000	625
Ref. S12			
Device area: $0.6 \times 1.2 \text{ mm}^2$			
Channel length and width: $100 \times 500 \mu\text{m}^2$			

P3HT: PEO/PS-PMMA-PS/ EMMI-TFSI Ref. S13 Channel length: 300 nm	Presynaptic spike: / -1 (V_D : 20)		0.00123
WSe ₂ /PEO: LiClO ₄ Ref. S14 Source–drain distance: $\approx 1 \mu\text{m}$	100	/	0.03
PETE-S/ETE-S: NaCl Ref. S15 Channel length and width: $30 \times 1000 \mu\text{m}^2$	V_G : 0.5 V_G : 20	/	1.1 1491
NaSbS ₂ Ref. S16 /	Constant 1000	bias: /	5.75
MXene/PEO: LiClO ₄ Ref. S17	80	6.4	~ 5.6
GDY/PEO: NaClO₄ (this work) Device area: $\pi/4 \times 0.33 \times 0.33 \text{ mm}^2$	5	0.0167	/

»

Page 41 (“Supplementary Information” part):

“References

- S1 Yan, X. *et al.* Memristor with Ag-Cluster-Doped TiO₂ Films as Artificial Synapse for Neuroinspired Computing. *Adv. Funct. Mater.* **28**, 1705320 (2018).
- S2 Sun, J. *et al.* Optoelectronic Synapse Based on IGZO-Alkylated Graphene Oxide Hybrid Structure. *Adv. Funct. Mater.* **28**, 1804397 (2018).
- S3 Gong, J. *et al.* Lateral Artificial Synapses on Hybrid Perovskite Platelets with Modulated Neuroplasticity. *Adv. Funct. Mater.* **30**, 2005413 (2020).
- S4 John, R. A. *et al.* Synergistic Gating of Electro-Iono-Photoactive 2D Chalcogenide Neuristors: Coexistence of Hebbian and Homeostatic Synaptic Metaplasticity. *Adv. Mater.* **30**, 1800220 (2018).
- S5 Yang, J. T. *et al.* Artificial Synapses Emulated by an Electrolyte-Gated Tungsten-Oxide Transistor. *Adv. Mater.* **30**, e1801548 (2018).
- S6 Yan, X. *et al.* Self-Assembled Networked PbS Distribution Quantum Dots for Resistive Switching and Artificial Synapse Performance Boost of Memristors. *Adv. Mater.* **31**, 1805284 (2019).
- S7 John, R. A. *et al.* Ionotronic Halide Perovskite Drift-Diffusive Synapses for Low-Power Neuromorphic Computation. *Adv. Mater.* **30**, e1805454 (2018).

- S8 Tan, H. *et al.* Broadband optoelectronic synaptic devices based on silicon nanocrystals for neuromorphic computing. *Nano Energy* **52**, 422-430 (2018).
- S9 Karbalaei Akbari, M. & Zhuiykov, S. A bioinspired optoelectronically engineered artificial neurobotics device with sensorimotor functionalities. *Nat. Commun.* **10**, 3873 (2019).
- S10 Zhu, L. Q., Wan, C. J., Guo, L. Q., Shi, Y. & Wan, Q. Artificial synapse network on inorganic proton conductor for neuromorphic systems. *Nat. Commun.* **5**, 3158 (2014).
- S11 van de Burgt, Y. *et al.* A non-volatile organic electrochemical device as a low-voltage artificial synapse for neuromorphic computing. *Nat. Mater.* **16**, 414-418 (2017).
- S12 Yao, X. *et al.* Protonic solid-state electrochemical synapse for physical neural networks. *Nat. Commun.* **11**, 3134 (2020).
- S13 Xu, W., Min, S.-Y., Hwang, H. & Lee, T.-W. Organic core-sheath nanowire artificial synapses with femtojoule energy consumption. *Sci. Adv.* **2**, e1501326 (2016).
- S14 Zhu, J. *et al.* Ion Gated Synaptic Transistors Based on 2D van der Waals Crystals with Tunable Diffusive Dynamics. *Adv. Mater.* **30**, e1800195 (2018).
- S15 Gerasimov, J. Y. *et al.* An Evolvable Organic Electrochemical Transistor for Neuromorphic Applications. *Adv. Sci.* **6**, 1801339 (2019).
- S16 Harikesh, P. C. *et al.* Cubic NaSbS₂ as an Ionic–Electronic Coupled Semiconductor for Switchable Photovoltaic and Neuromorphic Device Applications. *Adv. Mater.* **32**, 1906976 (2020).
- S17 Wei, H. *et al.* Redox MXene Artificial Synapse with Bidirectional Plasticity and Hypersensitive Responsibility. *Adv. Funct. Mater.* 2007232, doi: 10.1002/adfm.202007232 (2020).”

Reviewer #3 (Remarks to the Author):

In this paper, the Authors investigate new materials for artificial synapses. They designed Graphdiyne-based artificial synapse for mimicking biological synapses and short-term plasticity. This paper is well written and clear to read. Results and applications of these new artificial synapses are impressive. However, the main utility of these artificial synapses is not clear. Is it for neuromorphic systems or for bio-hybrid systems or for both? Could these artificial synapses be embedded in a neuromorphic system? And what is the maximum frequency these artificial synapses can work?

RESPONSE:

We would like to sincerely thank the reviewer for the very positive comments and constructive suggestions. We have revised the manuscript and added some discussion and results (new Supplementary Figs. 11) according to your instructive suggestions/comments and believe that these revisions have improved the paper. We have tried to explain the specific issues point-to-point in the following parts.

Please find below our responses (in blue) to each of your specific comments (in black). Revisions to the original article are indicated in red.

1. In the introduction part, it could be nice to add few sentences to explain in which systems these artificial synapses can be added.

L56 It could improve the state of the art of the paper adding these references on bio-hybrid systems:

Bucci, S. et al. A Neuromorphic Prosthesis to Restore Communication in Neuronal Networks. *iScience* 19, 402–414 (2019).

Keren, H., Partzsch, J., Marom, S. & Mayr, C. G. A Biohybrid Setup for Coupling Biological and Neuromorphic Neural Networks. *Front. Neurosci.* 13, 432 (2019).

Serb, A. et al. Memristive synapses connect brain and silicon spiking neurons. *Sci. Rep.* 10, 1–7 (2020).

Mosbacher, Y. et al. Toward neuroprosthetic real-time communication from in silico to biological neuronal network via patterned optogenetic stimulation. *Sci. Rep.* 10, 7512 (2020).

L60 It could improve the state of the art of the paper adding these references on microfluidic synapses and artificial synapses:

Keene, S. et al. A biohybrid synapse with neurotransmitter-mediated plasticity, *Nature Materials*, 19:969-973 (2020).

Levi, T. et Fujii, T. Microfluidic neurons: a new way in neuromorphic engineering?, *Micromachines*, 7:146 (2016).

RESPONSE:

Thank you very much for your helpful suggestions. We have modified the “Introduction” part and added some statements on bio-hybrid systems to clarify importance and the novelty of our topic. We have cited the references as suggested, with some corresponding changes in the revised manuscript.

Revised parts:

We have revised and added the relevant information to the revised manuscript as follows.

Page 3 (“Introduction” part):

“Complex human nervous systems, which have the edges of being highly compact, parallel, and reliable, are gaining more and more attention in various fields such as neuromorphic computing, bioinspired sensory systems, **brain machine interfaces**, and prosthetics.”

Page 3 (“Introduction” part):

“Therefore, the imitation of synapses as building blocks for neural processing are of crucial importance for constructing an efficient **artificial sensorimotor system, and bio-hybrid system and neuromorphic chip.**”

Page 3 (“Introduction” part):

“Among them, the configurations featuring ion migrations can opportunely facilitate the neuromorphic synapses to emulate the biological sensory/motor neurons in bioinspired ionotronic systems and bio-hybrid systems.”

Page 3 (“Introduction” part):

“These intriguing ion shuttle characteristics inspire a new perspective of constructing GDY-based artificial synapses (GASs) for mimicking synaptic cleft information transmission, which is promising for plasticity-mediated signal processing and transmission in bio-hybrid systems and artificial sensorimotor systems.”

Page 4 (“Introduction” part):

“Inspired by biological motor neuron, an integrated ionic artificial efferent nerve can be constructed to mimic the real-time processing and manipulation of signals. Here, for the first time, a junction-type GAS has been proposed by coupling the GDY film with the solid-state electrolytes for emulating multiple short-term plasticity such as postsynaptic current, paired-pulse facilitation (PPF), and dynamic filtering, with outstanding pulse responsiveness and femtowatt-level energy consumptions. The GASs can retain good ion diffusion dynamics under relatively high temperature (~353 K) and humidity (~35%). Attempts to exploit the short-term plasticity of GDY in bioinspired analogous efferent nerve have demonstrated the real-time information integration, the parallel processing capabilities, and the signal transduction and actuation, and therefore paved the way to future bioinspired ionotronic sensorimotor systems.”

Page 22 (“References” part):

7. Buccelli, S. *et al.* A Neuromorphic Prosthesis to Restore Communication in Neuronal Networks. *iScience* **19**, 402-414 (2019).
8. Keren, H., Partzsch, J., Marom, S. & Mayr, C. G. A Biohybrid Setup for Coupling Biological and Neuromorphic Neural Networks. *Front. Neurosci.* **13**, 432 (2019).
9. Serb, A. *et al.* Memristive synapses connect brain and silicon spiking neurons. *Sci. Rep.* **10**, 2590 (2020).
10. Mosbacher, Y. *et al.* Toward neuroprosthetic real-time communication from in silico to biological neuronal network via patterned optogenetic stimulation. *Sci. Rep.* **10**, 7512 (2020).
18. Keene, S. T. *et al.* A biohybrid synapse with neurotransmitter-mediated plasticity. *Nat. Mater.* **19**, 969-973 (2020).
19. Levi, T. & Fujii, T. Microfluidic Neurons, a New Way in Neuromorphic Engineering? *Micromachines* **7**, 146 (2016).”

2. It is not clear for me if you can use your artificial synapse in two mode: excitatory and inhibitory. I can see the shunting inhibition but could you also provide negative current? As I

explained in my general comment, which will be the best system that you can design using your artificial synapses? Neuromorphic chip? Bio-hybrid system? Both?

It could be interesting to use your synapses with the most famous neuromorphic chip like Indiveri group, Spinnaker, Loihi, etc.) or even more biomimetic artificial neurons like in Khoiratee et al. work. Here are some references on these neuromorphic chips.

S. Furber, D. Lester, L. Plana, J. Garside, E. Painkras, S. Temple, A. Brown, Overview of the SpiNNaker system architecture, *IEEE Transactions on Computers*, 62:12, 2454-2467, 2012;

P. Merolla et al., A million spiking-neuron integrated circuit with a scalable communication network and interface, *Science*, 345:6197, August 2014;

M. Davies et al., Loihi: A neuromorphic manycore processor with on-chip learning. *IEEE Micro*, 38(1):82–99, Jan 2018;

G. Indiveri et al., Neuromorphic silicon neuron circuits, *Frontiers in Neuroscience*, 5:73, 2011;

F. Khoiratee et al., Optimized real-time biomimetic neural network on FPGA for bio-hybridization, *Frontiers in Neuroscience*, 13-377, 2019.

RESPONSE:

Thank you very much for your helpful comments and suggestions. At the same time, we would also like to thank you for your enthusiasm for giving us so many useful documents for reference, which we have included to enrich our reference list (Refs. 20-24).

For the current output by the device, it can be seen that the device outputs positive current and negative current under the action of positive and negative pulses from the I - V curve. Moreover, the post-synaptic currents triggered by positive and negative pulses show volatile characteristics, which is applicable to signal transduction rather than information storage.

For the best system using our artificial synapses, our original intention of designing this synaptic device is to apply it to the intelligent sensorimotor system for signal transduction. Furthermore, based on the many advantages of GDY itself, the volatility of alkali metal ion storage and its unique biocompatibility, the application of GDY in bio-hybrid systems is feasible, which may be our next main work. For applications on neuromorphic chips that combines computation and memory, the currently prepared GDY artificial synapse needs to be modified in structure for better long-term plasticity.

3. Could you describe more in details your experiments using artificial neurons, how do you connect your nA current to the other artificial device? Is it possible to design one system using the same substrate (artificial neurons and artificial synapses)? One other point is not clear. Which level of amplitude you need to get answer from your synapses? In the manuscript you explained several times that you using pulse at volt level (ex: +2 V, 440 ms). But at the end, you talked about mV pulse. Maybe I missed one explanation, but could you describe which exact level of pulse you need to activate the synapses?

RESPONSE:

Thank you very much for your sincere comments and suggestions. In the revised manuscript, we have described in more detail the experimental process of how to connect artificial neurons, and the operating parameters involved.

For connecting our nA current to the other artificial device, the postsynaptic current (~nA) is amplified and transduced to voltage to actuate the artificial muscle. Here, the voltage transduced by our artificial efferent nerve is about +3 V, which could be sufficient to drive the artificial muscles.

For a system using the same substrate, it may be feasible to construct an artificial neuron all based on GDY. We can replace the existing artificial muscles with GDY artificial muscles (Ref. 74) and connected them with GDY artificial synapses in the future.

For the operational amplitude level of the device, it can be observed from the I - V curve of our device that the operating voltage of our device can be in a wide range from +4 V to tens of millivolts or even millivolts. Hence, the millivolt level pulse is used for finding how low energy consumption per synaptic event can be achieved.

Revised parts:

We have revised and added the relevant information to the revised manuscript and Supplementary Information as follows.

Page 18 (“Discussion” part):

“Furthermore, it may be more interesting to construct an all-GDY efferent nerve by combining our GAS and GDY-based artificial muscles in the future.”

Page 40 (“Supplementary Information” part):

“

Supplementary Fig. 12 Diagram of synaptic device-amplifier circuit-polymer actuator system. As a demonstration, the nano-amp-level postsynaptic current of a single device (Na-GAS) is amplified and the motor neuron synaptic potential is output to drive the artificial muscle (polymer actuator). Here, an operational amplifier was introduced to output the desired voltage (+3 V) and operate the actuator. The amplifier circuit amplifies the input voltage by 250,000 times to reach the working voltage of the actuator of ~3 V. The bottom electrode of the synaptic device was connected to the amplifier circuit to convert currents (~12 nA) to output voltages, such that the

actuator can be operated. One end of the bottom electrode of the synaptic device was coated with silver paste and dried in air. Copper wires were used for a connection.”

4. In supplementary Figure 7, you show results using low frequency pulse. Usually in neuromorphic systems the spike activity frequency is higher. How your artificial synapses answer to higher pulse frequency (10-200Hz)?

RESPONSE:

Thank you very much for your helpful comments. We agree that the spike activity frequency in neuromorphic systems is important and synaptic response at a frequency of several hertz (5.6 Hz) was provided. In order to obtain synaptic responses at higher frequencies, the interval between adjacent pulses has been reduced from the initial few seconds (~3 s) to the current sub-second (~0.18 s). In the supplementary material, we have increased the pulse frequency from sub-hertz to 5.6 Hz (new Supplementary Figs. 11) and our GAS still shows good parallel processing capabilities and short-term plasticity. As for the response of GAS under higher frequency pulse input signals (10-200 Hz), no accurate conclusion has yet been obtained from our devices due to the limitation of the time accuracy of the test equipment. However, the synaptic response at higher frequencies is still of our concern, and we will further explore this phenomenon in the near future after upgrading of our testing equipment.

Revised parts:

We have added the relevant information to the “Supplementary Information” as below.

Page 39 (“Supplementary Information” part):

“

Supplementary Fig. 11 a Postsynaptic current triggered by one or two presynaptic inputs at 2.34 Hz. b Postsynaptic current triggered by one or two presynaptic inputs with different frequency (2.34 and 4.28 Hz). c Postsynaptic current triggered by one or two presynaptic inputs at 5.6 Hz. Here, as the pulse frequency continues to increase, the synapse output terminal is still sensitive to the pulse timing of the two input terminals, that is, when the applied two sets of pulses overlap in time (Supplementary Fig. 11a and c), the synapse weight will double. If these two sets of pulse frequencies are different, this gain effect can still be observed (Supplementary Fig. 11b).”

5. You described results using artificial muscles. Is your system biocompatible (as Graphdiyne is biocompatible)? If yes, could you culture some neurons on your synapses and test this bio-hybrid system? For instance, preneuron or postneuron are biological neurons. Could you add this study

of feasibility on the discussion part?

RESPONSE:

Thank you very much for your instructive comments and opinions. The current system, an efferent neuron for the intelligent motor system, is not biocompatible due to the existence of incompatible components. However, we will cultivate some neurons as our presynaptic neurons in the next work for a series of attempts and explorations. Therefore, we believe that further optimization of the structure of Li-GAS and Na-GAS is inevitable. It is possible to design a Graphdiyne synaptic device to connected with biological neurons (Ref. 9 and Ref. 18) to construct a bio-hybrid system for spike transmission and plasticity emulation as a future work. As the reviewer commented, Graphdiyne is biocompatible by itself, so we think it will be more interesting to implement the release of neurotransmitters (such as dopamine) to realize the transduction of bio-electrochemical signals. Finally, we made a brief feasibility discussion of the GDY-based bio-hybrid system in the “Discussion” part of the revised manuscript.

Revised parts:

We have added the relevant information to the revised manuscript as below.

Page 18 (“Discussion” part):

“In addition, GDY has outstanding biological activity due to its active acetylene unit, which has already emerged in the fields of biosensing, drug delivery and living micromotors. Hence, GDY can be coupled with biological presynaptic neurons viably to form a bio-artificial hybrid system due to its biocompatibility to complete the spike transmission and plasticity processing. A functional biohybrid GDY system, a bio-electrochemical signal input terminal and neuromorphic GAS output terminal, can be conceived to demonstrate the regulation of synaptic weights in neuromorphic-based prosthetics.”

Page 22 (“References” part):

- “9. Serb, A. *et al.* Memristive synapses connect brain and silicon spiking neurons. *Sci. Rep.* **10**, 2590 (2020).
46. Yuan, K., Asunción-Nadal, V., Li, Y., Jurado-Sánchez, B. & Escarpa, A. Graphdiyne Micromotors in Living Biomed. *Chem. Eur. J.* **26**, 8471-8477 (2020).
72. Wu, L. *et al.* Graphdiyne: A new promising member of 2D all-carbon nanomaterial as robust electrochemical enzyme biosensor platform. *Carbon* **156**, 568-575 (2020).
73. Jin, J. *et al.* Graphdiyne Nanosheet-Based Drug Delivery Platform for Photothermal/Chemotherapy Combination Treatment of Cancer. *ACS Appl. Mater. Interfaces* **10**, 8436-8442 (2018).”

6. Could you compare our artificial synapses with other artificial synapses like Memristor? Advantages and drawbacks.

RESPONSE:

Thank you very much for your helpful suggestions.

The low-voltage operating characteristics of electrolyte-based devices provide the possibility for the implementation of ultralow energy consumption artificial synapse. Our GAS also exhibits ultralow voltage response and sub-biological power consumption, which is competitive with other two-terminal resistive switches (new Supplementary Table 1) but is still in a dilemma in the large-scale preparation of single- and multi-layer GDY. The application of electrolyte-based devices has been studied in brain-like computers and artificial intelligent systems but fundamental researches on device scalability and durability still need further efforts. Network level demonstration with these electrolyte-based devices is needed in the future study.

We have tried to compare the advantages and drawbacks of neuromorphic devices like memristors with our devices in the revised manuscript “Discussion” section added a table (new Supplementary Table 1) to compare the pulse amplitude, power and energy consumption, and device specifications of the last findings about the power/energy consumption of synaptic devices in the revised “Supplementary Information” part.

Revised parts:

We have added the relevant information to the revised manuscript and Supplementary Information as follows.

Page 18 (“Discussion” part):

“Furthermore, GAS exhibits ultralow voltage response and sub-biological power consumption, which is competitive with other two-terminal resistive switches (e.g. memristors) but is still in a dilemma in the large-scale preparation of single- and multi-layer GDY.”

Page 32 (“Supplementary Information” part):

“**Supplementary Table 1.** Comparison of pulse amplitude, power and energy consumption of artificial synapse devices.

Device structures	Pulse amplitude / mV	Power consumption / pW	Energy consumption / pJ
Ag-Cluster-Doped TiO ₂ Ref. S1 Memristor size: $\pi \times 50 \times 50 \mu\text{m}^2$	±1000	/	SET: 26.0 RESET: 22.9
IGZO/Alkylated Graphene Oxide Ref. S2 Channel length and width: $10 \times 10 \mu\text{m}^2$	-500	/	Capacitive: 136 Resistive: 14.3
CH ₃ NH ₃ PbBr ₃ Single Crystalline Ref. S3 Channel length and width: $100 \times 260 \mu\text{m}^2$	-30	0.0157	0.0143
MoS ₂ /DEME-TFSI Ref. S4	2000 50000	/	Ionotronic: 4.8 Electronic: 13000

Channel length and width: $9 \times 20 \mu\text{m}^2$			
WO ₃ /DEME-TFSI	V _G : 600	519	36
Ref. S5	V _{SD} : 300		
Channel size: $500 \times 50 \mu\text{m}^2$			
PbS Quantum Dots/Ga ₂ O ₃	SET: 120~260	SET: ~1000	/
Ref. S6	RESET: -50~-190	RESET: ~1000000	
Memristor size: $\pi \times 50 \times 50 \mu\text{m}^2$			
FAPbBr ₃	/	/	2300
Ref. S7			
Device area: 0.1 mm^2			
Silicon Nanocrystals	/	/	0.7
Ref. S8	(Optical Pulse)		
Device area: $2 \times 2 \text{ mm}^2$			
In-Doped TiO ₂	/	/	2.41
Ref. S9	(Optical Pulse)		
100 μm gap (source-drain)			
IZO/Nanogranular SiO ₂	Spike amplitude: /		15
Ref. S10	300		
Channel thickness: 20 nm	(Gate bias: -700)		
PEDOT: PSS/Nafion/	/	/	~10
PEDOT: PSS-PEI			
Ref. S11			
Device area: $10^3 \mu\text{m}^2$			
WO ₃ /Nafion-117 resin	250	125000	625
Ref. S12			
Device area: $0.6 \times 1.2 \text{ mm}^2$			
Channel length and width: $100 \times 500 \mu\text{m}^2$			
P3HT: PEO/PS-PMMA-PS/	Presynaptic spike: /		0.00123
EMMI-TFSI	-1		
Ref. S13	(V _D : 20)		
Channel length: 300 nm			
WSe ₂ /PEO: LiClO ₄	100	/	0.03
Ref. S14			
Source-drain distance: $\approx 1 \mu\text{m}$			
PETE-S/ETE-S: NaCl	V _G : 0.5	/	1.1
Ref. S15	V _G : 20		1491
Channel length and width: $30 \times 1000 \mu\text{m}^2$			
NaSbS ₂	Constant	bias: /	5.75

Ref. S16	1000		
/			
MXene/PEO: LiClO ₄	80	6.4	~5.6
Ref. S17			
GDY/PEO: NaClO₄ (this work)	5	0.0167	/
Device area: $\pi/4 \times 0.33 \times 0.33 \text{ mm}^2$			

Page 41 (“Supplementary Information” part):

“References

- S1 Yan, X. *et al.* Memristor with Ag-Cluster-Doped TiO₂ Films as Artificial Synapse for Neuroinspired Computing. *Adv. Funct. Mater.* **28**, 1705320 (2018).
- S2 Sun, J. *et al.* Optoelectronic Synapse Based on IGZO-Alkylated Graphene Oxide Hybrid Structure. *Adv. Funct. Mater.* **28**, 1804397 (2018).
- S3 Gong, J. *et al.* Lateral Artificial Synapses on Hybrid Perovskite Platelets with Modulated Neuroplasticity. *Adv. Funct. Mater.* **30**, 2005413 (2020).
- S4 John, R. A. *et al.* Synergistic Gating of Electro-Iono-Photoactive 2D Chalcogenide Neuristors: Coexistence of Hebbian and Homeostatic Synaptic Metaplasticity. *Adv. Mater.* **30**, 1800220 (2018).
- S5 Yang, J. T. *et al.* Artificial Synapses Emulated by an Electrolyte-Gated Tungsten-Oxide Transistor. *Adv. Mater.* **30**, e1801548 (2018).
- S6 Yan, X. *et al.* Self-Assembled Networked PbS Distribution Quantum Dots for Resistive Switching and Artificial Synapse Performance Boost of Memristors. *Adv. Mater.* **31**, 1805284 (2019).
- S7 John, R. A. *et al.* Ionotronic Halide Perovskite Drift-Diffusive Synapses for Low-Power Neuromorphic Computation. *Adv. Mater.* **30**, e1805454 (2018).
- S8 Tan, H. *et al.* Broadband optoelectronic synaptic devices based on silicon nanocrystals for neuromorphic computing. *Nano Energy* **52**, 422-430 (2018).
- S9 Karbalaee Akbari, M. & Zhuiykov, S. A bioinspired optoelectronically engineered artificial neurobotics device with sensorimotor functionalities. *Nat. Commun.* **10**, 3873 (2019).
- S10 Zhu, L. Q., Wan, C. J., Guo, L. Q., Shi, Y. & Wan, Q. Artificial synapse network on inorganic proton conductor for neuromorphic systems. *Nat. Commun.* **5**, 3158 (2014).
- S11 van de Burgt, Y. *et al.* A non-volatile organic electrochemical device as a low-voltage artificial synapse for neuromorphic computing. *Nat. Mater.* **16**, 414-418 (2017).
- S12 Yao, X. *et al.* Protonic solid-state electrochemical synapse for physical neural networks. *Nat. Commun.* **11**, 3134 (2020).

- S13 Xu, W., Min, S.-Y., Hwang, H. & Lee, T.-W. Organic core-sheath nanowire artificial synapses with femtojoule energy consumption. *Sci. Adv.* **2**, e1501326 (2016).
- S14 Zhu, J. *et al.* Ion Gated Synaptic Transistors Based on 2D van der Waals Crystals with Tunable Diffusive Dynamics. *Adv. Mater.* **30**, e1800195 (2018).
- S15 Gerasimov, J. Y. *et al.* An Evolvable Organic Electrochemical Transistor for Neuromorphic Applications. *Adv. Sci.* **6**, 1801339 (2019).
- S16 Harikesh, P. C. *et al.* Cubic NaSbS₂ as an Ionic–Electronic Coupled Semiconductor for Switchable Photovoltaic and Neuromorphic Device Applications. *Adv. Mater.* **32**, 1906976 (2020).
- S17 Wei, H. *et al.* Redox MXene Artificial Synapse with Bidirectional Plasticity and Hypersensitive Responsibility. *Adv. Funct. Mater.* 2007232, doi: 10.1002/adfm.202007232 (2020).”

REVIEWERS' COMMENTS

Reviewer #1 (Remarks to the Author):

The manuscript has been adequately revised. I have no further comments.

Reviewer #2 (Remarks to the Author):

Although some improvements have been made in the revised version of manuscript entitled "Graphdiyne-based Ultra-Responsive Artificial Synapse with multi-Ion diffusive dynamics for Mimicking efferent Nerves", in my view the quality of English is far from accepting level for NC. It is very difficult to have a grasp of the logic of the experiments and results.

Authors added several new measurements to the main text and to the supplementary file. The results now look more authentic and valuable for early-stage evaluation of synaptic behavior of Graphdiyne-based materials. The present style of the manuscript can reflect more ion-diffusive dynamics in this type of material than the other discussed and claimed factors.

However, authors claimed "proof-of-concept" for artificial efferent nerves to implement the integrated output of multiple synaptic inputs despite the fact that the proof-of-concept for artificial afferent nerves has already been reported (Science 360 (2018) 998-1008). Thus, I am still not convinced that the presented research is the GROUND-BRAKING research.

If the manuscript is going to be accepted, I would strongly suggest to the authors to seek professional help from the English native speakers in order to improve presentation of the results. Manuscript in its current form is very difficult to read.

Reviewer #3 (Remarks to the Author):

Thanks for this improved document. Authors answered to most of my questions and comments.

Reviewers' Comments:

Reviewer #1 (Remarks to the Author):

The manuscript has been adequately revised. I have no further comments.

Reviewer #2 (Remarks to the Author):

Although some improvements have been made in the revised version of manuscript entitled "Graphdiyne-based Ultra-Responsive Artificial Synapse with multi-Ion diffusive dynamics for Mimicking efferent Nerves", in my view the quality of English is far from accepting level for NC. It is very difficult to have a grasp of the logic of the experiments and results.

Authors added several new measurements to the main text and to the supplementary file. The results now look more authentic and valuable for early-stage evaluation of synaptic behavior of Graphdiyne-based materials. The present style of the manuscript can reflect more ion-diffusive dynamics in this type of material than the other discussed and claimed factors.

However, authors claimed "proof-of-concept" for artificial efferent nerves to implement the integrated output of multiple synaptic inputs despite the fact that the proof-of-concept for artificial afferent nerves has already been reported (Science 360 (2018) 998-1008). Thus, I am still not convinced that the presented research is the GROUND-BREAKING research. If the manuscript is going to be accepted, I would strongly suggest to the authors to seek professional help from the English native speakers in order to improve presentation of the results. Manuscript in its current form is very difficult to read.

Reviewer #3 (Remarks to the Author):

Thanks for this improved document. Authors answered to most of my questions and comments.

Point-to-Point Responses to Reviewers' Comments

Reviewer #1 (Remarks to the Author):

The manuscript has been adequately revised. I have no further comments.

RESPONSE:

We appreciate Reviewer 1 for the favorable comments.

Reviewer #2 (Remarks to the Author):

Overall comments:

Although some improvements have been made in the revised version of manuscript entitled “Graphdiyne-based Ultra-Responsive Artificial Synapse with multi-Ion diffusive dynamics for Mimicking efferent Nerves”, in my view the quality of English is far from accepting level for NC. It is very difficult to have a grasp of the logic of the experiments and results.

Authors added several new measurements to the main text and to the supplementary file. The results now look more authentic and valuable for early-stage evaluation of synaptic behavior of Graphdiyne-based materials. The present style of the manuscript can reflect more ion-diffusive dynamics in this type of material than the other discussed and claimed factors.

However, authors claimed "proof-of-concept" for artificial efferent nerves to implement the integrated output of multiple synaptic inputs despite the fact that the proof-of-concept for artificial afferent nerves has already been reported (Science 360 (2018) 998-1008). Thus, I am still not convinced that the presented research is the GROUND-BRAKING research.

If the manuscript is going to be accepted, I would strongly suggest to the authors to seek professional help from the English native speakers in order to improve presentation of the results. Manuscript in its current form is very difficult to read.

RESPONSE:

We appreciate Reviewer 2 for the criticism and the helpful suggestions. We have tried our best to improve the quality of English of the manuscript. In particular, we have sought the help from Springer Nature Author Services with the presentation of our research results. We have also removed the statement about “proof-of-concept” in the manuscript. We believe the revised and improved manuscript is now acceptable for publication in *Nature Communications*.

Reviewer #3 (Remarks to the Author):

Thanks for this improved document. Authors answered to most of my questions and comments.

RESPONSE:

We appreciate Reviewer 3 for the favorable comments.